# Analysis of common differential gene expression between rheumatoid arthritis and ulcerative colitis

**Peng-fei Han**[1‡], **Wei-rong Cui**[2‡], **Fang-zheng He**[1,2], **Tao Wu**[1,2], **Chang-sheng Liao**[1*]

**1** Department of Orthopaedics, Heping Hospital Affiliated to Changzhi Medical College, Changzhi, P.R. China, **2** Department of Graduate School, Graduate Student Department of Changzhi Medical College, Changzhi, P.R. China

‡ PH and WC are considered as co-first authors.
* 15603441409@163.com

## Abstract

### Objective

This study employs bioinformatics analysis with the objective of identifying commonly differentially expressed genes (DEGs) in ulcerative colitis (UC) and rheumatoid arthritis (RA), as well as exploring their underlying molecular mechanisms. By doing so, it aims to provide a theoretical basis for investigating the potential associations between these two diseases and developing novel therapeutic strategies.

### Materials and methods

We downloaded multiple gene expression datasets for Rheumatoid Arthritis (RA) and Ulcerative Colitis (UC) from the Gene Expression Omnibus (GEO) database. For RA, GSE77298, GSE12021, and GSE55457 were selected as the training sets, with GSE89408 serving as the validation set. For UC, GSE36807, GSE87473, and GSE92415 were chosen as the training sets, and GSE13367 as the validation set. During data processing, we first merged the RA and UC data from each training set with standardized data, eliminated batch effects, and obtained combined datasets of differentially expressed genes (DEGs). Subsequently, we conducted a cross-analysis of the DEGs from RA and UC to identify commonly up-regulated and down-regulated genes. To gain a deeper understanding of these DEGs, we constructed a protein-protein interaction (PPI) network and identified hub genes. For further analysis of these hub genes, we utilized the GENEMANIA platform to obtain functional annotations and interaction information. Finally, we validated our analysis results using the GSE89408 and GSE13367 datasets.

**Data availability statement:** The relevant code has been uploaded to 10.6084/m9.figshare.30639671.

**Funding:** The author(s) received no specific funding for this work.

**Competing interests:** The authors have declared that no competing interests exist.

## Results

After a thorough analysis of the differentially expressed genes in the cells of patients with rheumatoid arthritis (RA) and ulcerative colitis (UC) we found that genes such as CCR7, CD19, CXCL13, CXCR4, and SELL were significantly up-regulated, suggesting their crucial roles in the pathology of both diseases. This discovery not only underscores the importance of these genes as biomarkers for the differential diagnosis of RA and UC, but also highlights key nodes worthy of further validation. In the future, it may be possible to slow or halt disease progression by modulating the expression of these genes.

## Conclusion

The results of this study reveal potential common molecular mechanisms underlying rheumatoid arthritis (RA) and ulcerative colitis (UC). The key target genes CCR7, CD19, CXCL13, CXCR4, and SELL highlight common underlying factors associated with both diseases. Further investigation and exploration of these findings can pave the way for new candidate targets and directions in therapeutic research aimed at treating RA and UC. This study emphasizes the importance of utilizing bioinformatics approaches to uncover the mechanisms of complex diseases, providing a promising pathway for the development of more effective and targeted treatments.

## 1. Introduction

Rheumatoid Arthritis (RA) and Ulcerative Colitis (UC) are two typical and prevalent autoimmune diseases that have long been the focus of medical research and clinical treatment [1,2]. These diseases not only severely impact patients' quality of life but also pose challenges for treatment due to their complex pathogenesis and diverse clinical manifestations [3]. Rheumatoid Arthritis is a chronic, systemic autoimmune disease that primarily affects the synovial membrane of joints, leading to joint inflammation, pain, swelling, and dysfunction [4,5]. Its pathogenesis involves abnormal activation of various immune cells and autoimmune reactions, resulting in persistent inflammation and destruction of the synovial membrane [6]. Inflammatory bowel disease (IBD) encompasses two subtypes: ulcerative colitis (UC) and Crohn's disease (CD). Unlike CD, which can affect the entire gastrointestinal tract, the inflammation in UC is confined to the colonic mucosa. Epidemiological data indicate that UC is more prevalent than CD globally [7]. Therefore, our study primarily focuses on exploring the association between rheumatoid arthritis (RA) and UC. As a chronic, non-specific inflammatory bowel disorder, UC predominantly affects the mucous membrane of the large intestine, leading to symptoms such as diarrhea, abdominal pain, and mucopurulent bloody stools. The exact etiology of UC remains incompletely understood, but it is currently believed to result from a combination of multiple factors, including genetic predisposition, environmental triggers, immune dysregulation, and alterations in the gut microbiota [8,9].

Research has demonstrated that patients with ulcerative colitis (UC) face a significantly higher risk of developing rheumatoid arthritis (RA) compared to those with other intestinal diseases. During the course of UC, some patients may exhibit extraintestinal manifestations such as arthritis, which is commonly referred to as "enteropathic arthritis" or "arthritis associated with ulcerative colitis" [10]. Despite the distinct anatomical sites and clinical presentations of RA and UC, studies have confirmed that both diseases involve characteristic abnormal activation of the immune system and chronic inflammatory responses (e.g., Fcγ receptor signaling). Additionally, there is significant genetic overlap between RA and UC in the IL-23/Th17 pathway, suggesting that they may share common genetic and molecular mechanisms [11–13]. Therefore, joint research on RA and UC is of paramount importance, as it can enhance diagnostic accuracy, optimize treatment strategies, and provide more effective therapeutic approaches for patients.

Bioinformatics, as an interdisciplinary field, provides powerful support for elucidating the genetic and molecular mechanisms of complex diseases by integrating knowledge and technologies from biology, computer science, and statistics [14]. In recent years, with the rapid development of bioinformatics techniques, in-depth analysis of gene expression differences between RA and UC has become feasible, providing a powerful tool for revealing their pathogenesis and identifying new therapeutic targets. The aim of this study is to use bioinformatics methods to conduct a joint research on Rheumatoid Arthritis and Ulcerative Colitis, revealing their differences and similarities at the gene expression level, identifying new biomarkers and potential disease-related genes, and deepening our understanding of RA and UC. These findings will not only contribute to a deeper understanding of the pathogenesis of RA and UC but may also provide new ideas for early diagnosis, prognosis assessment, and formulation of treatment strategies for these diseases.

In summary, this study will utilize bioinformatics methods to comprehensively analyze the gene expression data of RA and UC, aiming to reveal their differences and similarities at the gene expression level and provide new perspectives and clues for in-depth research and treatment of these diseases.

## 2. Materials and methods

**2.1. Data Acquisition:** In this study, we primarily utilized microarray technology for gene expression analysis and obtained datasets related to Rheumatoid Arthritis (RA) and Ulcerative Colitis (UC) from the Gene Expression Omnibus (GEO) public database.,We selected datasets with a sample size greater than 20 to ensure sufficient statistical power for reliably identifying differentially expressed genes (DEGs). Only datasets with clearly defined disease groups and control groups were included to avoid phenotypic confounding.We searched and extracted datasets using the following keywords: "Rheumatoid Arthritis," "Ulcerative Colitis," "microarray," "human samples," and corresponding disease-specific gene expression patterns. The datasets included in our analysis were RA (GSE77298, GSE12021, and GSE55457 as the training set, with GSE89408 used as the validation dataset) and UC (GSE36807, GSE87473, and GSE92415 as the training set, with GSE13367 used as the validation set). The GSE77298 (RA) dataset contains 23 samples, including 16 disease samples and 7 control samples; GSE12021 (RA) contains 21 samples, with 12 disease samples and 9 control samples; GSE55457 (RA) contains 23 samples, with 13 RA samples and 10 control samples; GSE36807 (UC) contains 22 samples, including 15 disease samples and 7 control samples; GSE87473 (UC) contains 127 samples, with 106 disease samples and 21 control samples; GSE92415 (UC) comprises 183 samples collected before (n = 87) and after (n = 75) treatment. It was originally generated to evaluate the efficacy of golimumab (GLM) induction therapy in moderate-to-severe ulcerative colitis. For our analysis, we selected the 87 pre-treatment UC samples together with 21 healthy control samples. Detailed age group and gender information were not available for all samples due to limitations in the data collection process. We acknowledge that parameters such as age, gender, treatment conditions, and comorbidities may exert potential confounding effects on gene expression profiles and serve as important background variables in certain analyses. However, our primary objective is to minimize batch effects across datasets through rigorous normalization procedures, thereby identifying differentially expressed genes (DEGs) associated with rheumatoid arthritis and ulcerative colitis, and exploring how these genes contribute to the progression of these diseases. Therefore, we have decided to include the

aforementioned datasets in our study.Detailed sample information and dataset characteristics for the RA and UC cohorts are summarized in S1 and S2 Tables, respectively.

**2.2. Data Preprocessing and DEG Identification:** We separately performed preprocessing on six gene expression datasets. Specifically, GSE77298, GSE12021, and GSE55457 were utilized for the analysis of rheumatoid arthritis, whereas GSE36807, GSE87473, and GSE92415 were employed for the analysis of ulcerative colitis. To ensure the quality of the data, we excluded genes (rows) or samples (columns) with a relatively high proportion of missing values. In cases where multiple probes corresponded to the same gene within a dataset, we calculated the arithmetic mean of their expression levels. This approach was taken to eliminate redundancy and streamline the subsequent analysis procedures. Subsequently, the processed expression matrices were normalized to rectify experimental batch effects and other systematic biases, thereby ensuring comparability across different datasets. Based on these preprocessed data, we utilized the limma package, which runs within the RStudio environment, to conduct differential expression analyses on the RA and UC groups in comparison to the normal control group. The limma package is well-suited for handling high-throughput expression data generated by microarray and RNA sequencing technologies. It can effectively estimate the fold change in gene expression and compute its statistical significance. We established stringent screening criteria: an adjusted P-value (adj.P.Val) $< 0.05$ and $|\log_2$ fold change (FC)$| > 1$. By setting these criteria, we were able to control the false discovery rate and simultaneously identify differentially expressed genes that were not only statistically significant but also held potential biological implications.

**2.3. Batch Effect Correction and Differential Analysis:** To mitigate batch effects that may arise from different experimental platforms, this study employed batch correction algorithms. We merged the standardized expression datasets from RA (GSE77298, GSE12021, and GSE55457) and UC (GSE36807, GSE87473, and GSE92415) respectively, and utilized the ComBat function provided by the sva package in RStudio to adjust for batch effects. The datasets generated before and after this normalization for RA and UC are available in S1–S4 Files.Subsequently, we used the ggpubr package to visualize the batch-corrected data. After correction, we conducted differential expression gene analysis using the limma package to identify differentially expressed genes with statistical significance in the merged datasets.

**2.4. Screening of Common DEGs in RA and UC:** After batch effect correction and differential expression analysis, DEGs in the RA and UC datasets were identified. A cross-analysis of the DEG sets for the two diseases enabled us to successfully identify common DEGs, including both upregulated and downregulated genes.

**2.5. Construction and Analysis of Protein-Protein Interaction (PPI) Network and Selection of Hub Genes:** Using the STRING online platform, we constructed a human PPI network containing 88 common DEGs from Rheumatoid Arthritis (RA) and Ulcerative Colitis (UC), focusing on association data for Homo sapiens. To ensure the credibility of protein interactions within the network, we established a minimum interaction score threshold of 0.400. Subsequently, the network was analyzed using the CytoHubba plugin in Cytoscape software (version 3.10.2) to identify hub genes. This analysis employed four different centrality algorithms—Degree, Closeness Centrality, Betweenness Centrality, and Maximal Clique Centrality(MCC)—to effectively detect genes playing a central role in the network. This multifaceted approach allowed us to identify hub genes that have a significant impact on the network's structure and function, circumventing the limitations of relying on a single network feature for gene identification and ensuring that the identified hub genes were validated in terms of their importance in the network from multiple perspectives.

**2.6. GENEMANIA Online Analysis:** To further investigate the interactions and functional associations between the ten hub genes selected in this study, we used the advanced gene function prediction tool GENEMANIA. By entering the names of these ten hub genes on the GENEMANIA website and selecting the appropriate species "Homo sapiens," an interaction network diagram was automatically constructed based on their known interactions and functional associations.

**2.7. Independent Validation of DEGs Using Additional Datasets:** During the validation phase, we utilized the public GSE89408 dataset for RA validation and the GSE13367 dataset for UC validation. Prior to conducting differential expression analysis, necessary preprocessing steps were performed, including normalization, to minimize technical variations

and ensure data comparability. The differential expression analysis aimed to identify significantly differentially expressed genes (DEGs) between the control group and each disease-specific cohort (RA and UC). Subsequently, we specifically examined the overlap between the DEGs identified in these datasets and the 10 hub genes previously identified in our preliminary analysis. To systematically evaluate the diagnostic potential of these 10 hub genes, we further performed receiver operating characteristic (ROC) curve analysis in the validation sets to quantify their ability to distinguish disease states. Meanwhile, leave-one-out cross-validation was employed to assess the robustness of the classification model constructed based on these genes.

**2.8. Characterization of immune cell infiltration:** To assess the potential influence of tissue heterogeneity and immune cell infiltration on hub genes, we performed quantitative immune cell infiltration analysis on RA and UC data using the CIBERSORT algorithm with its LM22 signature matrix. Based on linear support vector regression, this method resolves the relative proportions of 22 immune cell subsets. To further distinguish whether hub gene expression is driven by immune cell abundance or stems from intrinsic tissue pathological responses, we calculated Pearson correlation coefficients between each hub gene's expression level and all immune cell proportions, accompanied by significance testing. This analysis aims to biologically differentiate genes directly associated with immune infiltration from those potentially representing core genes involved in key disease pathways.

**2.9. Subcellular Localization Analysis:** To gain deeper insights into the functional localization and potential mechanisms of the hub genes within cells, we performed a systematic subcellular localization analysis of the five selected hub genes (SELL, CCR7, CD19, CXCL13, and CXCR4). The relevant data were obtained from the Human Protein Atlas database, with batch downloading and processing conducted using the HPAanalyze package in RStudio. This database encompasses protein localization information for over 15,000 human genes. Based on the primary subcellular localization of their encoded proteins, these genes were categorized into the following four classes: membrane proteins (primarily localized to the plasma membrane), secreted proteins (mainly secreted into the extracellular space), cytoplasmic proteins (primarily localized in the cytoplasm), and nuclear proteins (primarily localized in the nucleus).

**2.10. Constructing the CeRNA Regulatory Network:** Potential interactions between core genes and lncRNAs (long non-coding RNAs) or miRNAs (microRNAs) were retrieved from four authoritative databases: miRanda, miRDB, miRTarBase, and TargetScan. These databases are highly reputable in the field of miRNA target prediction, each employing distinct algorithms and data sources to predict interactions between miRNAs and their target genes. The selection of these four databases as references was based on their ability to provide multi-dimensional data support, thereby enhancing the reliability and comprehensiveness of the prediction results.After obtaining predictions from these four databases, rigorous cross-validation was performed. Cross-validation is a critical bioinformatics analysis method used to evaluate the accuracy and consistency of predictions by comparing results derived from different data sources or algorithms. In this study, only lncRNA–miRNA pairs that were consistently recorded and mutually supported across all four databases—miRanda, miRDB, miRTarBase, and TargetScan—were retained, thereby strengthening the reliability and consistency of the predicted outcomes.Subsequently, an in-depth analysis was conducted to examine the interactions between these validated miRNAs and the hub genes. Based on the filtered interaction pairs, a ceRNA regulatory network was constructed using Cytoscape 3.10.2.

**2.11. Statistical Methods, Software, and Tools:** All statistical analyses were performed in the R environment. The two-sample t-test was used to analyze differences in gene expression between groups. The Benjamini-Hochberg method was employed for multiple testing correction to control the false discovery rate (FDR). Data processing and statistical analysis were conducted using R language (version 4.3.1).The intermediate analysis files and results generated during the above statistical procedures are available in S1 Data.

## 3. Results

**3.1. Identification of Differentially Expressed Genes (DEGs):** After analysis, a series of differentially expressed genes (DEGs) were identified across different datasets. Specifically, the GSE36807 dataset revealed 471 DEGs in the uc group,

the GSE87473 dataset showed 928 DEGs in the uc group, and the GSE92415 dataset indicated 1020 DEGs in the uc group. For the RA group, the GSE77298 dataset displayed 432 DEGs, the GSE12021 dataset exhibited 235 DEGs, and the GSE55457 dataset demonstrated 314 DEGs.

**3.2. Batch Correction and Differential Expression Analysis:** After standardization procedures, we initially integrated the annotated expression datasets for the RA group (GSE77298, GSE12021, and GSE55457) and the UC group (GSE36807, GSE87473, and GSE92415). Subsequently, we eliminated batch effects from the combined datasets. The subsequent differential expression analysis revealed that there were 416 DEGs in the RA group and 778 DEGs in the UC group. The complete lists of differentially expressed genes identified in the RA and UC analyses are provided in S5 and S6 Files, respectively (Figs 1–6).

**3.3. Selection of Commonly Expressed Genes in RA and UC:** We conducted differential expression analysis on the combined and batch-effect-corrected datasets for rheumatoid arthritis (RA) and ulcerative colitis (UC) groups, obtaining their respective differentially expressed genes (DEGs). By intersecting the two sets of DEGs, we precisely identified 77 upregulated genes and 11 downregulated genes. The upregulated genes include CXCL13, CXCL10, PSMB9, CXCL9, AIM2, IGHM, SNX10, BIRC3, IGLC1, SEMA4A, IGLJ3, KMO, CXCL11, CD72, GBP1, GZMH, TNFSF11, GZMK, TRBC1, PNOC, EVI2B, CCL18, PLA2G2D, CFB, CYTIP, CD79A, ITGB2, LAMP3, CXCR4, MS4A1, LAX1, WNT5A, ISG20, RASGRP1, IL7R, IGHG1, LCP1, FKBP11, RHOH, FAIM3, IGHD, CD19, MMP1, PIM2, LOXL1, RAC2, SLAMF7, SPP1, SPAG4, GZMB, KIAA0125, CHI3L2, TNIP3, CR2, PLA2G7, CORO1A, CXCL6, IGLV6–57, CD38, CCL19, SEMA4D, RGS1, CCR7, MMP3, TNFAIP6, ZBED2, AQP9, FCGR1B, P2RX5, SERPINA1, LILRA3, HS3ST3A1, BCL2A1, ICOS, MMP9, SELL, LTF. The 11 downregulated genes are AKR1B10, TRHDE, CYP4F12, PDE6A, MAOA, PCK1, NPY1R, EIF1AY, RPS4Y1, MB, and ADH1C (Fig 7).

**3.4. Construction of Protein-Protein Interaction (PPI) Network and Selection of Hub Gene:** In this study, a protein-protein interaction (PPI) network was constructed based on the common differentially expressed genes (DEGs) identified in rheumatoid arthritis (RA) and ulcerative colitis (UC). As shown in the following figure, the network comprises 88 shared DEGs, with disconnected nodes in the network hidden. It is represented by 88 nodes and 345 edges, where the nodes signify proteins encoded by DEGs and the edges represent interactions between these proteins. Utilizing

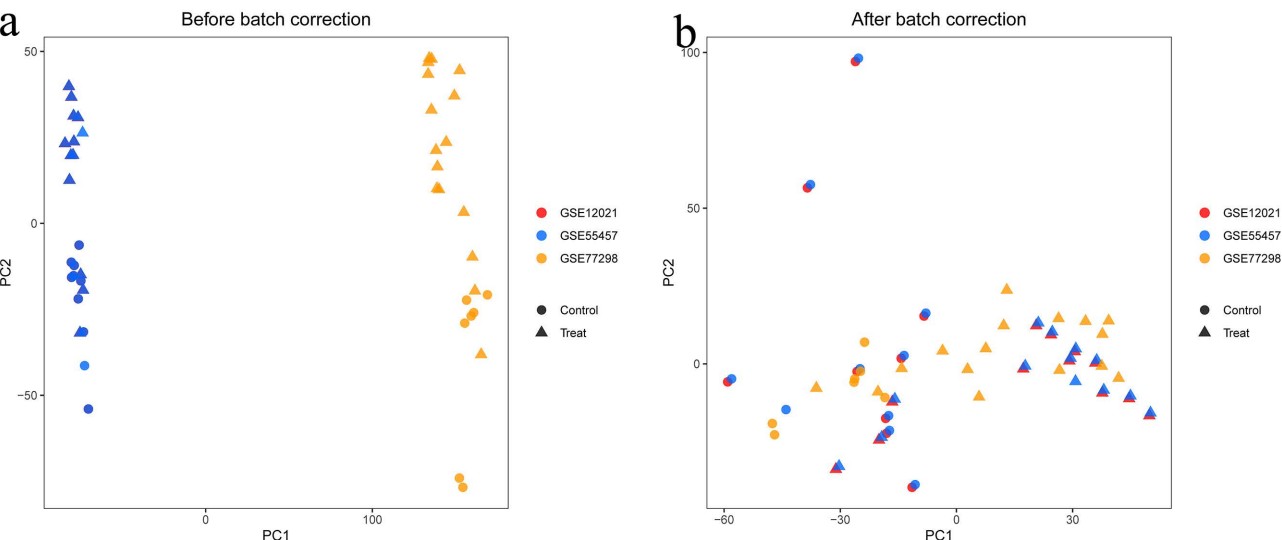

**Fig 1. Principal component analysis (PCA) of gene expression data before and after batch correction.** 1a represents the combined datasets of RA before batch correction. 1b displays the merged RA dataset post-batch correction.

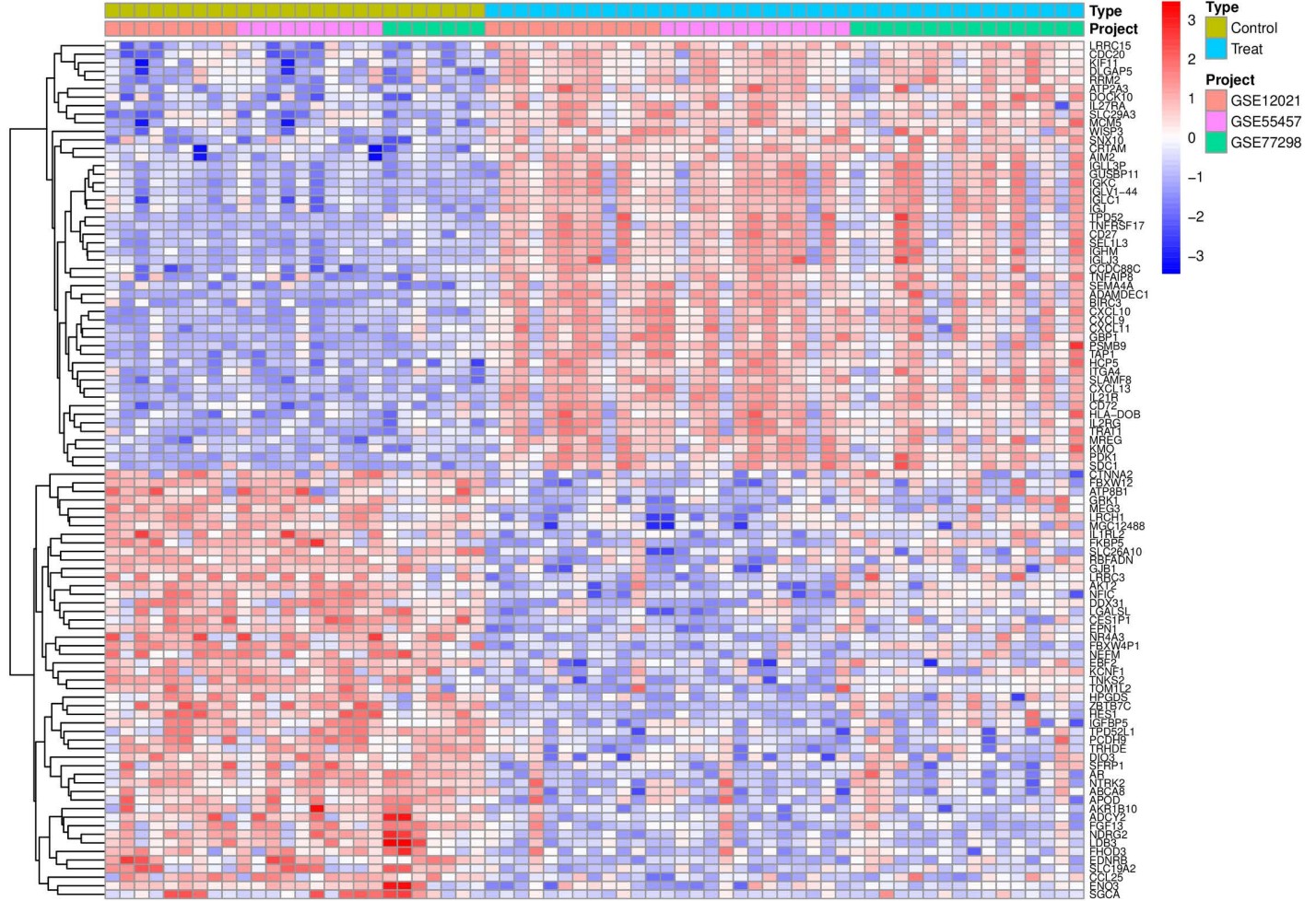

**Fig 2. Displays a heatmap, where green represents the normal control group, light blue represents the disease group, and pink, purple, and green distinctions indicate different datasets within the groups.** Red indicates high expression levels, while blue indicates low expression levels.

Cytoscape software and the Degree, Closeness Centrality, Betweenness Centrality, and Maximal Clique Centrality (MCC) algorithms provided by the cytohubba plugin, along with summing their scores, the top ten key hub genes were identified: CXCL13, CCR7, CCL19, CD19, GZMB, CXCL9, CD38, CXCR4, IL7R, and SELL. The complete lists of hub genes identified at confidence thresholds of 0.300, 0.400, and 0.500, as well as their intersection, are provided in S7–S10 Files, respectively.The positions of these hub genes within the network and the intensity of the node colors reflect their connectivity and importance within the network,Furthermore, ROC analysis demonstrated that these hub genes exhibit strong discriminatory power in distinguishing disease states. Leave-one-out cross-validation revealed highly consistent discriminatory performance of RA-associated hub genes across different datasets. The AUC values from the full dataset analysis and leave-one-out validation showed strong correlation (points closely distributed along the diagonal), indicating that the discriminatory capability of these genes is not dependent on specific datasets. For UC-associated hub genes, the leave-one-out validation results showed good reproducibility. Although the point distribution was relatively scattered, most genes maintained effective discriminatory ability (AUC > 0.8) when excluding any single dataset. Notably, the robustness of UC hub genes exhibited some heterogeneity: some genes remained significant in all three validation rounds, while others

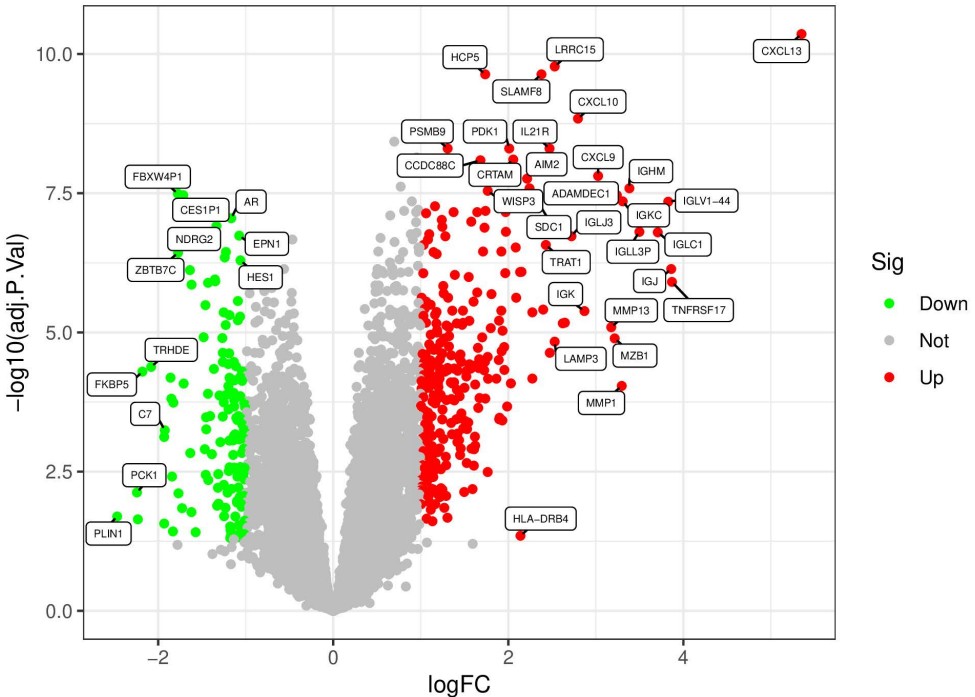

**Fig 3. Presents a volcano plot, with green symbolizing low expression and red symbolizing high expression.**

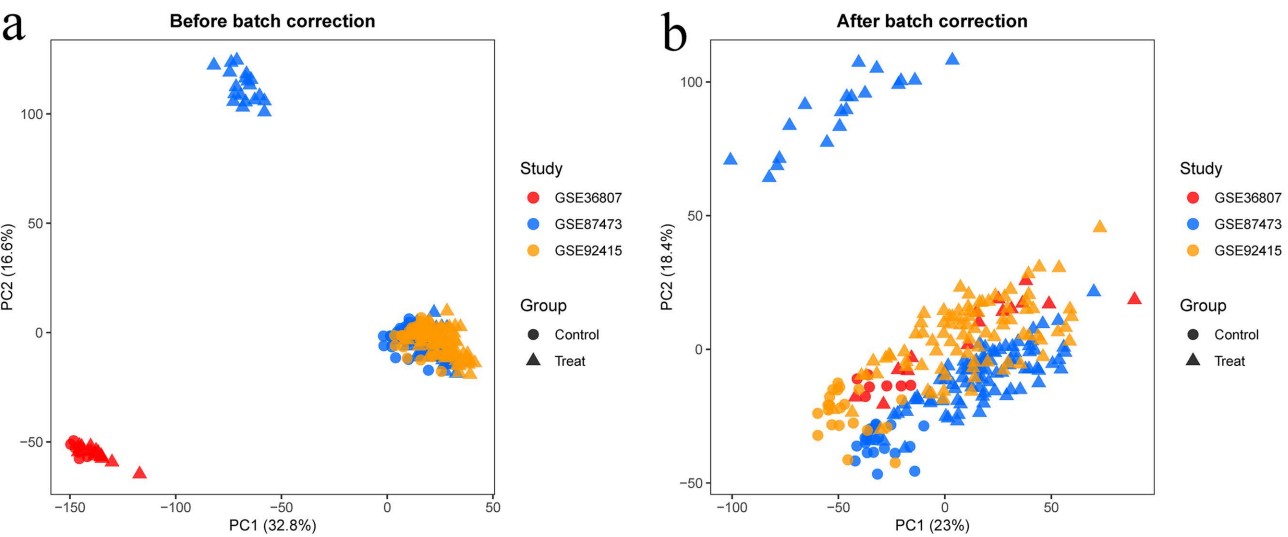

**Fig 4. Depicts the combined datasets of UC prior to batch correction. 2b displays the merged UC dataset post-batch correction.**

showed fluctuating significance across different data subsets. This variation may reflect the impact of UC disease heterogeneity on gene expression, yet the overall discriminatory performance of the core gene set was validated.The detailed results of the ROC analysis for hub genes in both rheumatoid arthritis and ulcerative colitis are provided in S11 and S12 Files, respectively (Figs 8–12).

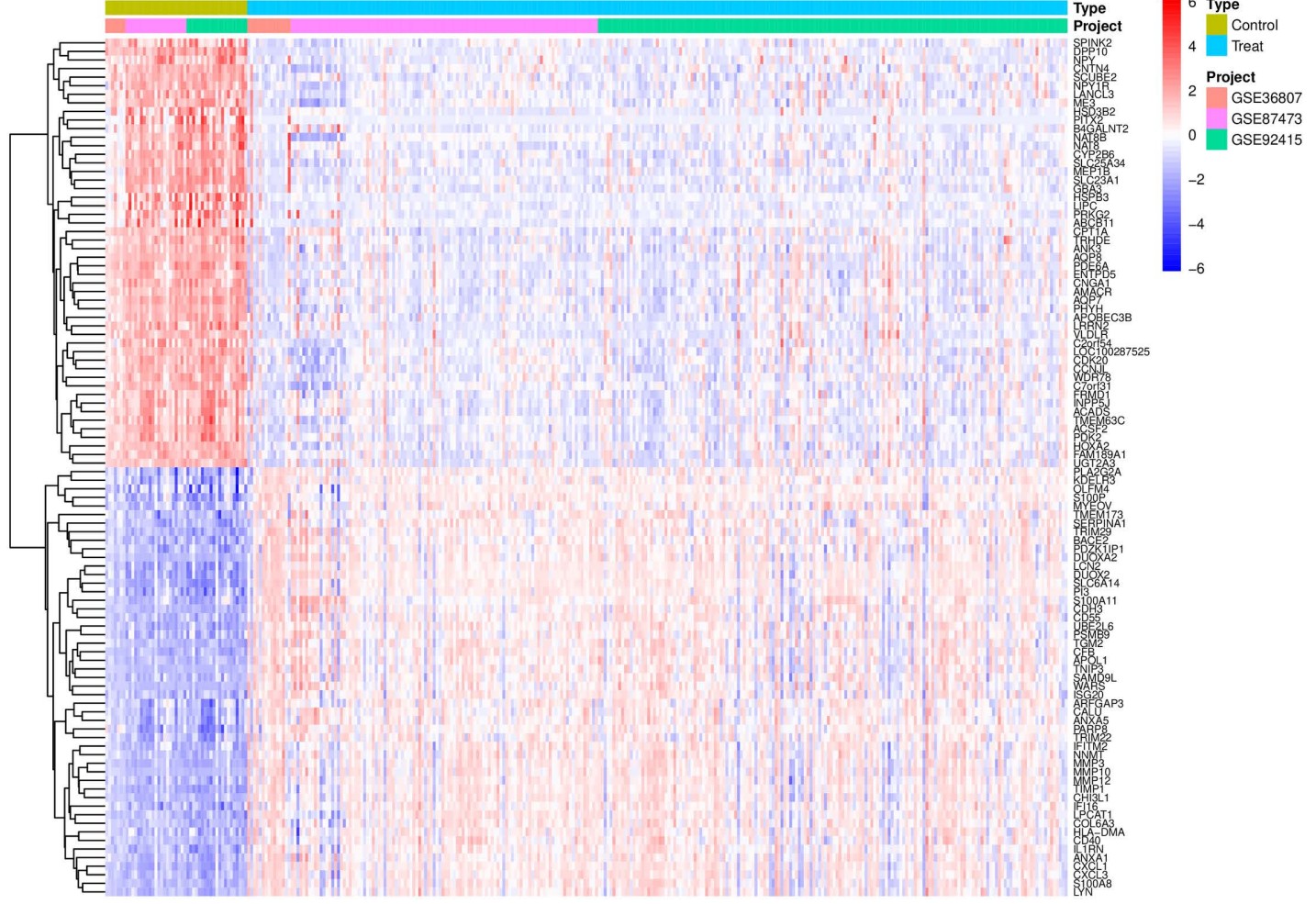

**Fig 5. Displays a heatmap, in which green represents the normal control group, light blue represents the disease group, and pink, purple, and green distinctions indicate the different datasets within these groups.** Red signifies high expression levels, while blue signifies low expression levels.

**3.5. GENEMANIA Online Analysis:** Analysis of the 10 hub differentially expressed genes (DEGs) (CXCL13, CCR7, CCL19, CD19, GZMB, CXCL9, CD38, CXCR4, IL7R, and SELL) on the GeneMANIA website revealed their central roles in various immune and inflammation-related biological processes. These genes are either directly involved or indirectly participate through interactions with other genes in immune cell activation, recruitment, and migration. The core processes identified by the analysis include leukocyte migration, leukocyte chemotaxis, cell chemotaxis, neutrophil migration, cellular response to biological stimuli, and cellular response to bacterial-derived molecules. This analysis underscores the pivotal roles of these hub genes in coordinating and driving immune cell migration, inflammatory responses, and responses to pathogen/damage signals. It suggests that the networks regulated by these genes represent potential mechanistic links between the pathophysiology of RA and UC, directly mediating tissue damage caused by immune cell infiltration under chronic inflammatory conditions. The extensive involvement of these genes in critical biological processes highlights their potential as therapeutic targets for modulating underlying disease processes and alleviating symptoms (Fig 13).

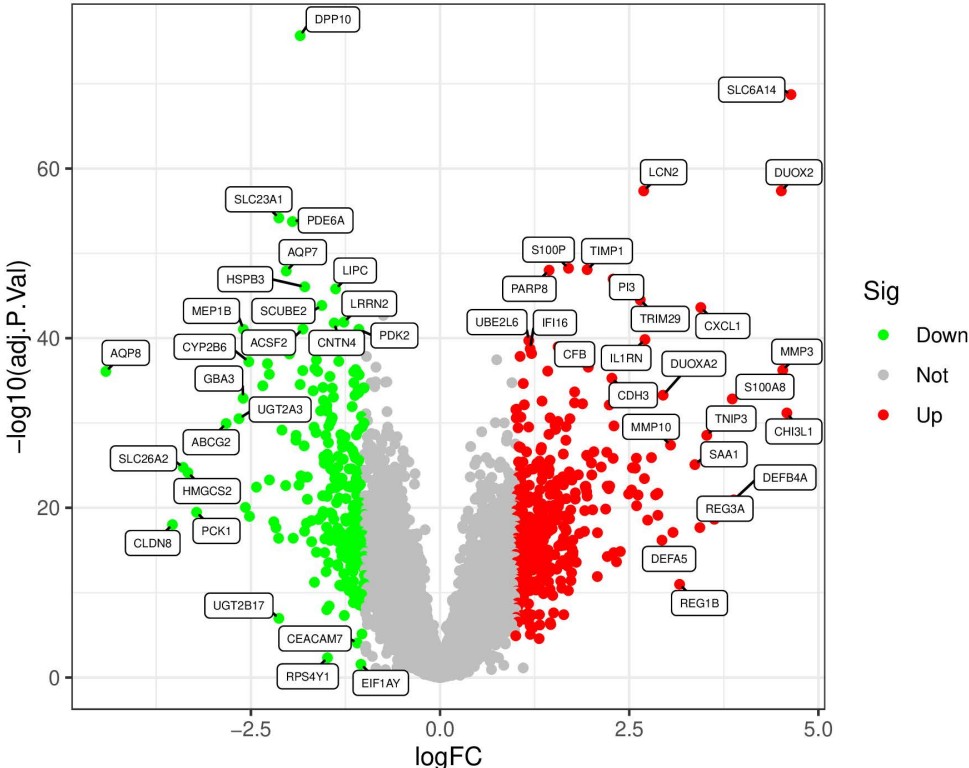

**Fig 6. Presents a volcano plot, with green indicating low expression and red indicating high expression.**

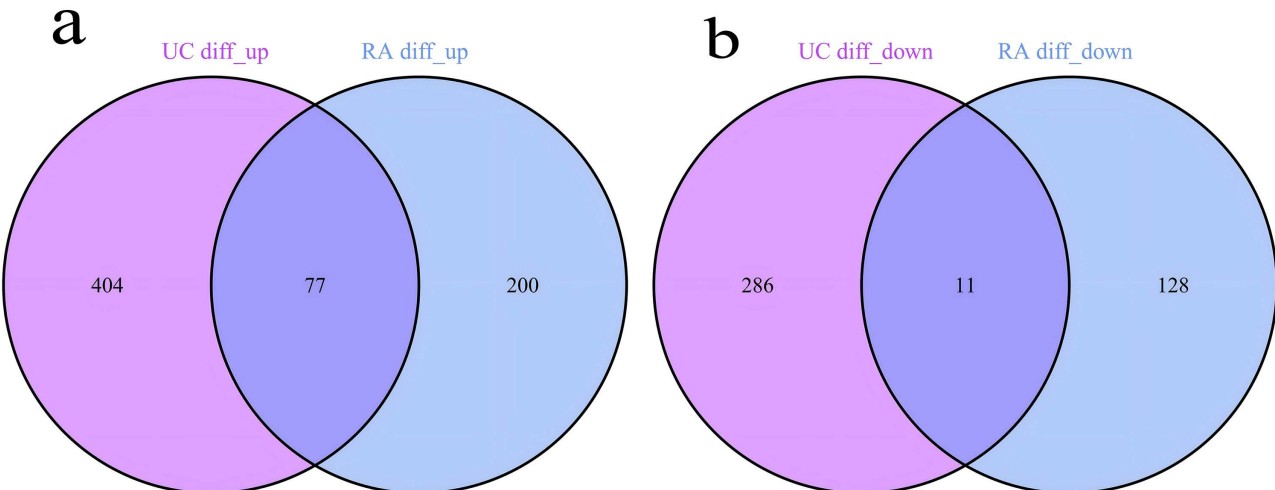

**Fig 7. Venn diagram of differentially expressed genes (DEGs) between rheumatoid arthritis (RA) and ulcerative colitis (UC).** 7a displays the 77 commonly upregulated differentially expressed genes (DEGs) identified in both RA and UC. 7b shows the 11 commonly downregulated DEGs identified in both RA and UC.

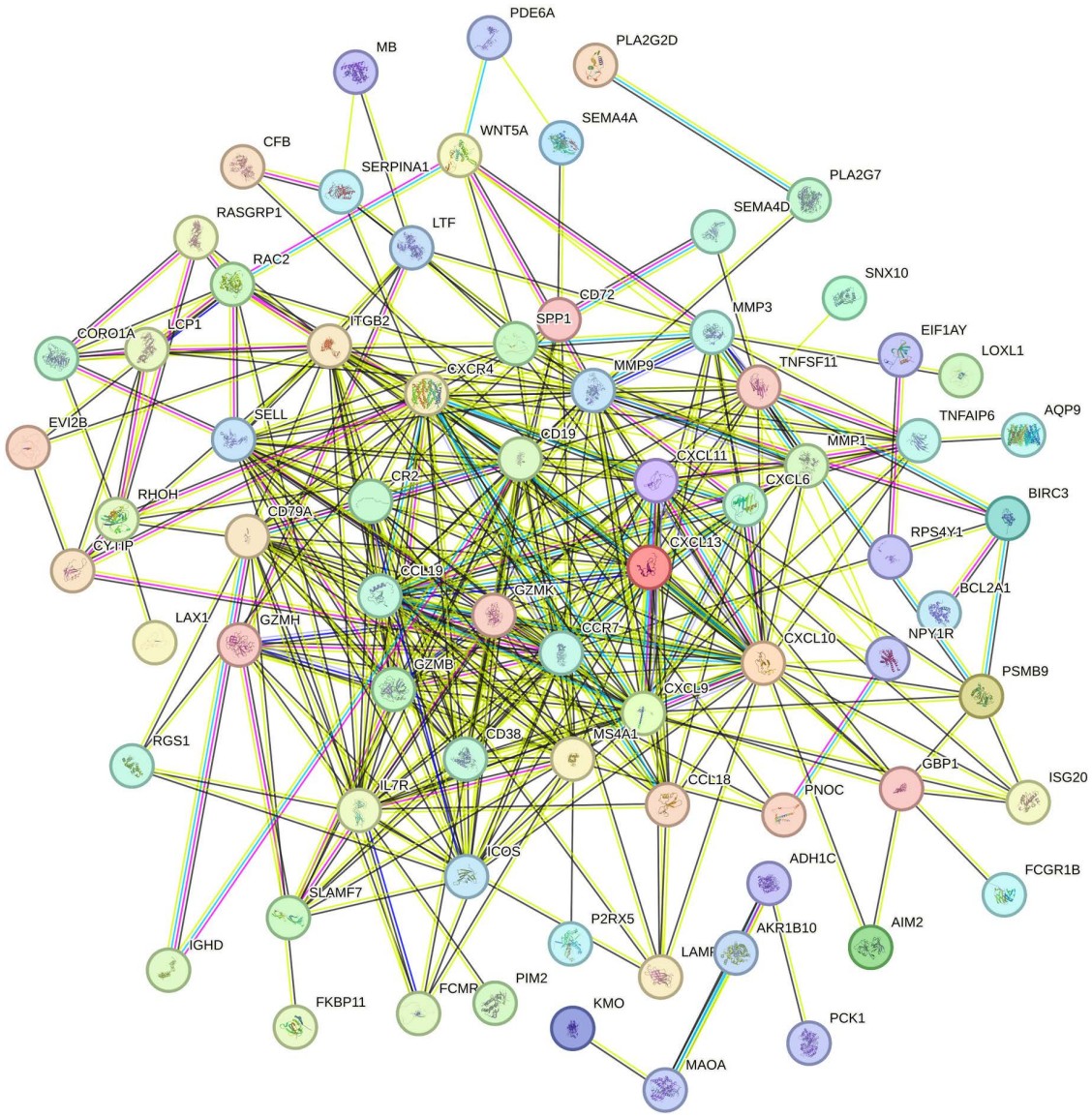

**Fig 8. Displays the protein-protein interaction (PPI) network generated by the 88 common differentially expressed genes (DEGs) identified between rheumatoid arthritis (RA) and ulcerative colitis (UC).**

**3.6. Independent Validation of DEGs Using Additional Datasets:** When performing validation analysis using the specified validation datasets, a threshold of $p < 0.01$ was selected to determine significant differences. This analysis aimed to validate the differential expression patterns of specific genes of interest compared to normal control groups in the context of RA and UC. The results of the validation analysis revealed significant differences in the expression levels of the CCR7, CD19, CXCL13, CXCR4, and SELL genes between the disease groups (RA and UC) and the normal control group. More specifically, compared to normal controls, the CCR7, CD19, CXCL13, CXCR4, and SELL genes exhibited significantly elevated expression levels in the RA and UC disease groups, indicating an upregulated state. The higher expression levels of these genes in RA and UC highlight their potential involvement in disease progression or inflammatory processes. The validation analysis emphasizes the importance of these genes as potential biomarkers or therapeutic

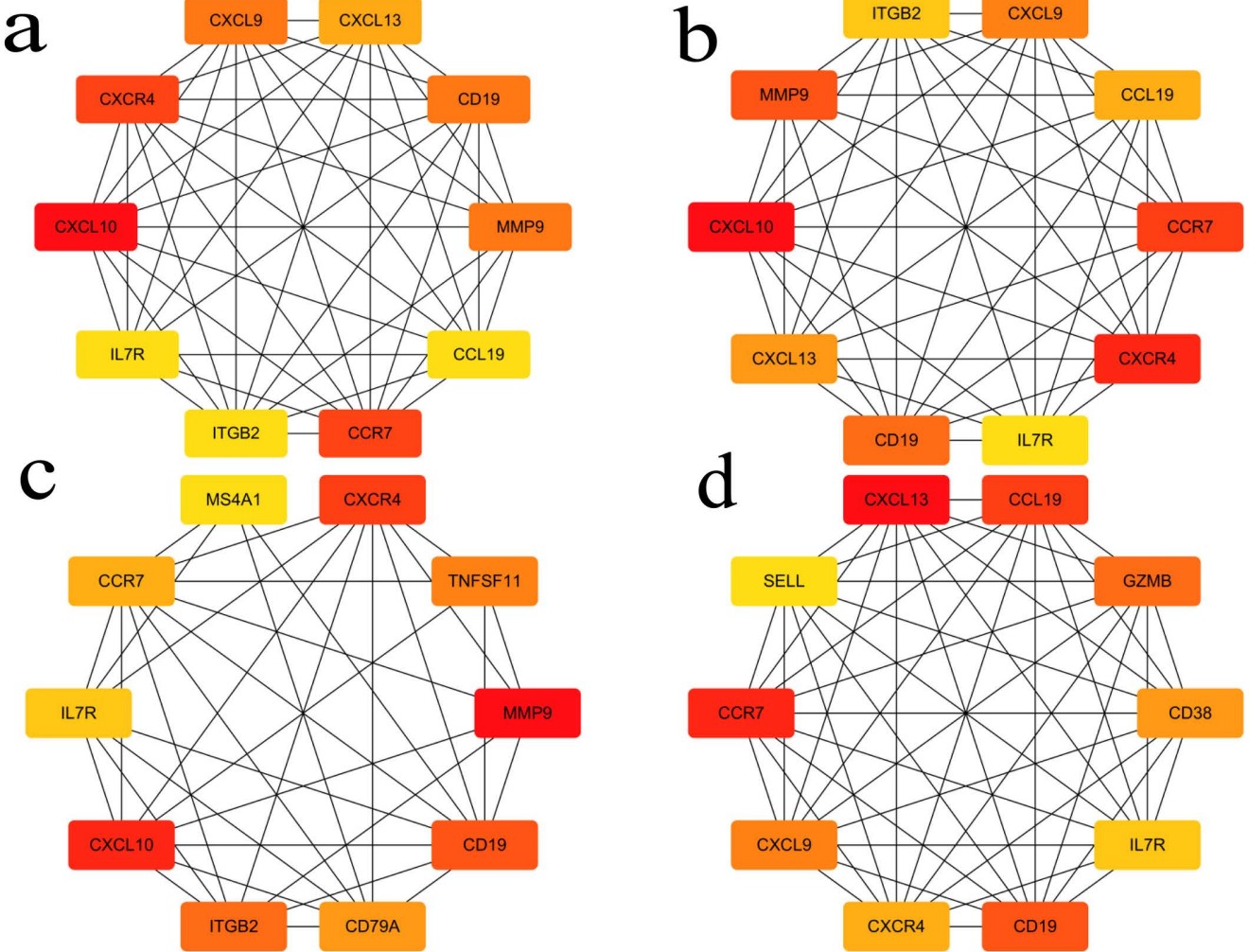

**Fig 9. Protein-protein interaction (PPI) networks of hub genes identified in rheumatoid arthritis (RA) and ulcerative colitis (UC).** 9a highlights the module of 10 identified central genes based on the degree metric. 9b showcases the closeness centrality of the genes. 9c illustrates the betweenness centrality of the genes. 9d presents a refined view of the Maximal Clique Centrality (MCC) of the genes.

targets for RA and UC. The upregulation of these genes suggests their role in disease exacerbation or as key players in potential pathogenic mechanisms, positioning them as possible targets for therapeutic interventions aimed at reducing their expression or activity in the context of the disease (Figs 14–18).

**3.7. Characterization of immune cell infiltration:** Given the confounding effects of immune cell infiltration,we systematically conducted immune infiltration association analyses for the five identified hub genes (SELL, CCR7, CD19, CXCL13, CXCR4) in both RA and UC cohorts. The results revealed distinct immune microenvironment patterns between the two diseases and clarified the potential roles of each hub gene within these contexts.In UC, CIBERSORT analysis demonstrated significant inter-sample heterogeneity in immune infiltration. Based on this, the five hub genes could be clearly categorized into two groups according to their correlations with immune cell abundance, reflecting their different functional emphases in the disease process.Genes closely associated with B-cell immune responses (CD19, CXCL13, SELL, CCR7):The expression of CD19 and CXCL13 showed highly significant positive correlations with infiltration

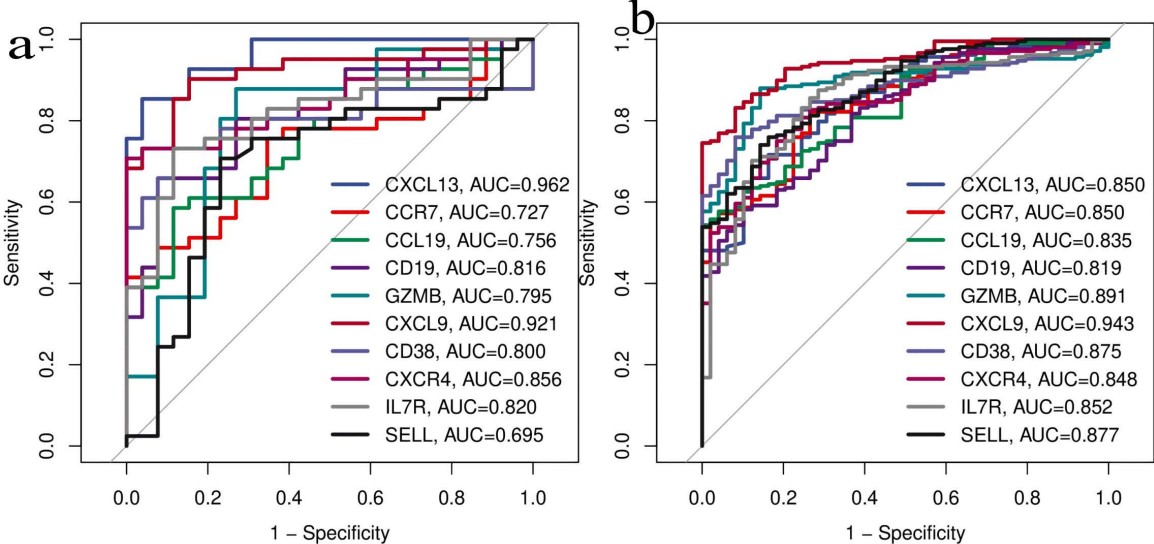

(PLOS One logo at top)

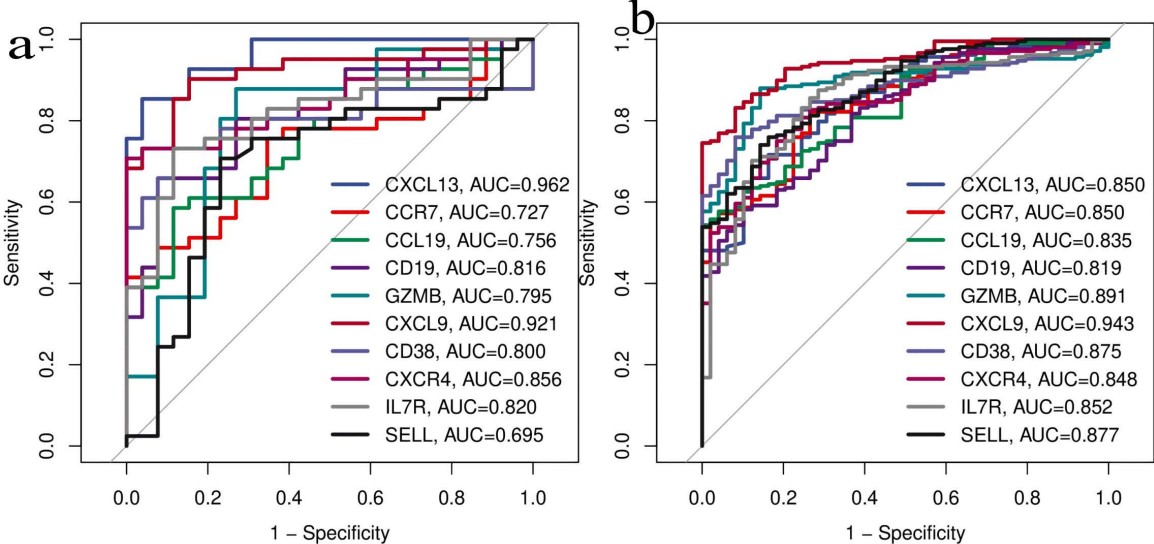

**Fig 10. Receiver operating characteristic (ROC) curves of hub genes in rheumatoid arthritis (RA) and ulcerative colitis (UC).** 10a:Validation of the Diagnostic Efficacy of Hub Genes in rheumatoid arthritis (RA).10b:Validation of the Diagnostic Efficacy of Hub Genes in ulcerative colitis (UC).

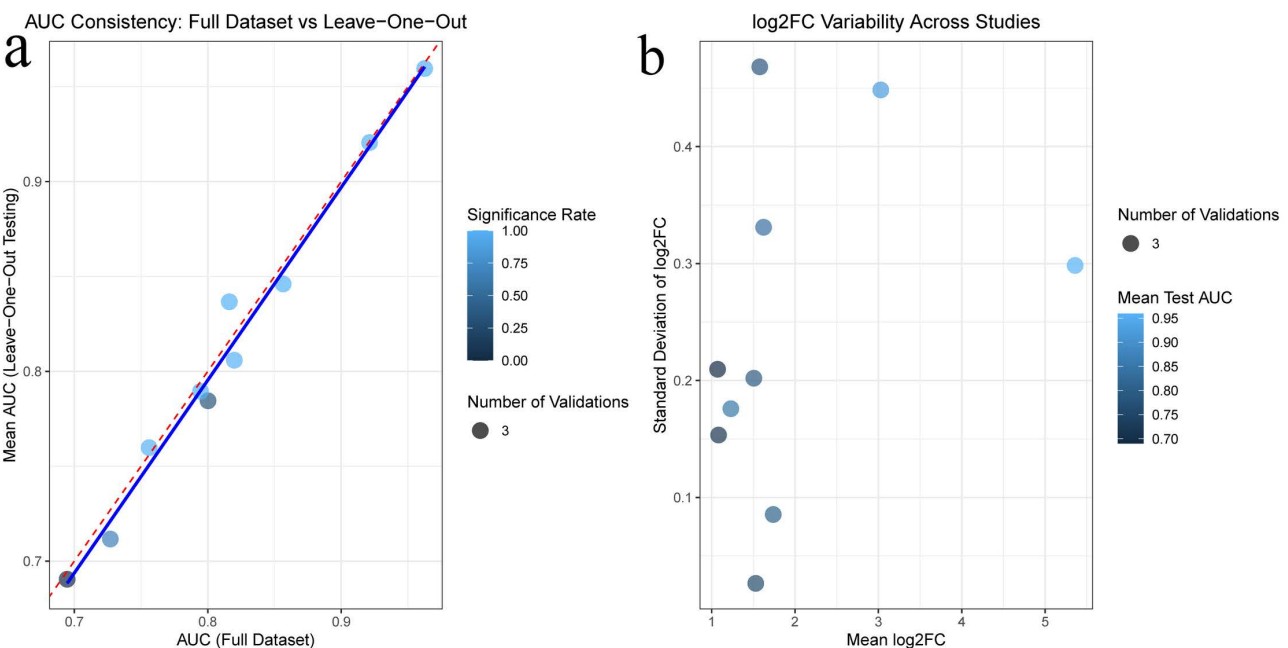

**Fig 11. Robustness validation of hub genes in Rheumatoid Arthritis 11a Correlation of Area Under the Curve (AUC) values obtained from the full dataset analysis and the leave-one-out cross-validation.** 11b Variability of gene expression changes (log2FC) across different datasets.

levels of B-cell lineages. For instance, CD19 correlated strongly with naive B cells (r = 0.599, p < 3.49e-26), plasma cells (r = −0.380, p < 3.30e-10), and follicular helper T cells (r = 0.513, p < 1.67e-18). CXCL13 was also significantly associated with memory B cells (r = 0.315, p < 2.83e-7) and follicular helper T cells (r = 0.480, p < 4.16e-16). These data strongly

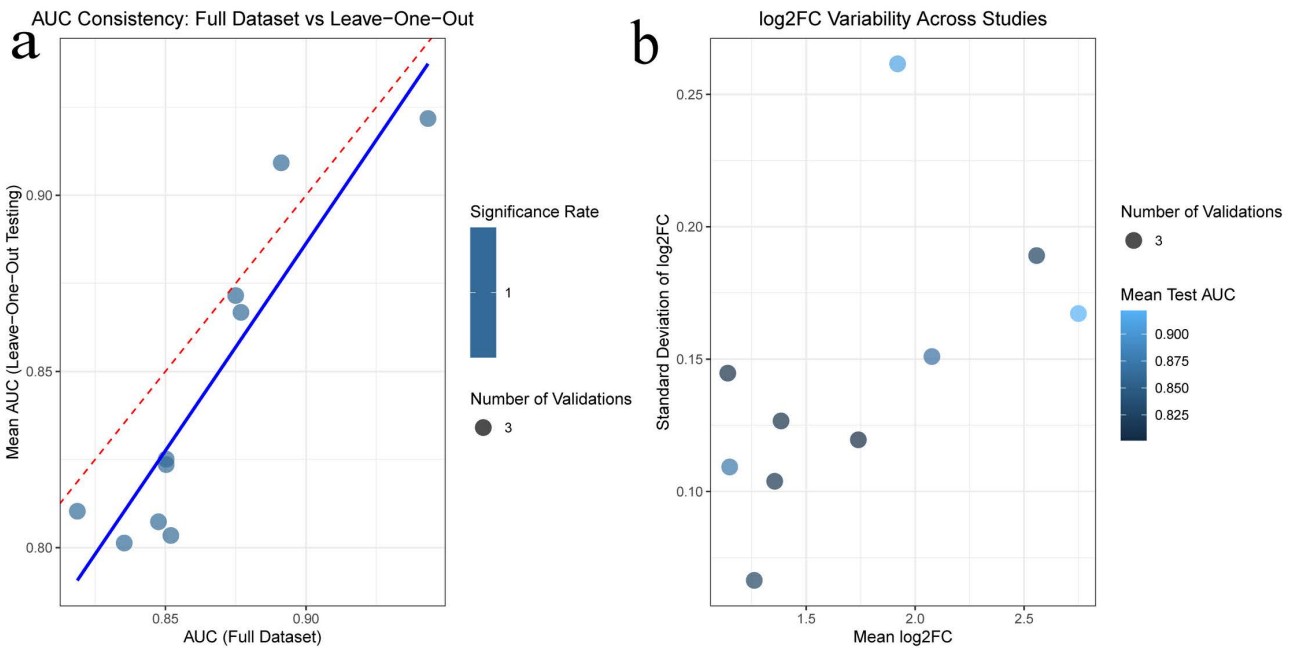

**Fig 12. Robustness validation of hub genes in Ulcerative Colitis 12a Correlation of Area Under the Curve (AUC) values obtained from the full dataset analysis and the leave-one-out cross-validation.** 12b Variability of gene expression changes (log2FC) across different datasets.

suggest that the hub status of CD19 and CXCL13 largely reflects the active aggregation of germinal center reactions and adaptive humoral immunity in UC mucosa.SELL and CCR7, key mediators of lymphocyte homing and migration, showed significant positive correlations with various immune cells (e.g., naive B cells, activated CD4 memory T cells, follicular helper T cells). However, they also exhibited strong negative correlations with M2 macrophages. This indicates that SELL and CCR7 not only participate in recruiting lymphocytes to intestinal inflammatory sites but also correlate with fluctuations in the immunosuppressive microenvironment, potentially playing a dual role in coordinating both the "advance" and "retreat" of immune responses.Gene relatively independent of immune infiltration (CXCR4):In stark contrast to the other four genes, CXCR4 expression showed no significant or only weak correlations with most immune cell subsets. Its correlations with all T-cell subsets, NK cells, and monocytes were non-significant ($p > 0.05$). The only notable correlation was a negative association with M2 macrophages ($r = -0.566$, $p < 5.15e-23$) and connections with other non-immune pathways. This crucial evidence suggests that CXCR4 may primarily participate in core pathological responses of intrinsic intestinal cells (such as epithelial or stromal cells), including cell migration and survival, with effects relatively independent of variations in immune cell numbers across samples.In RA, we observed a markedly different pattern: the immune infiltration profile of RA synovial tissue showed high consistency across samples, lacking the significant heterogeneity observed in UC. Accordingly, correlation analysis yielded a definitive conclusion: the expression levels of all five hub genes (SELL, CCR7, CD19, CXCL13, CXCR4) showed no statistically significant associations with the proportions of any of the 22 immune cell subsets analyzed (all p-values substantially $> 0.05$). Specifically, even genes highly correlated with immune cells in UC showed no significant correlations in RA. This result excludes confounding effects of inter-sample variations in immune cell composition on gene expression measurements.This key finding indicates that in RA, the expression changes of our identified hub genes are not driven by the abundance of infiltrating immune cells. Instead, they are more likely to represent core pathogenic molecular programs that are stably activated within intrinsic synovial tissue cells (such as fibroblast-like synoviocytes) during the disease state. Since the expression of these genes is unaffected by fluctuating

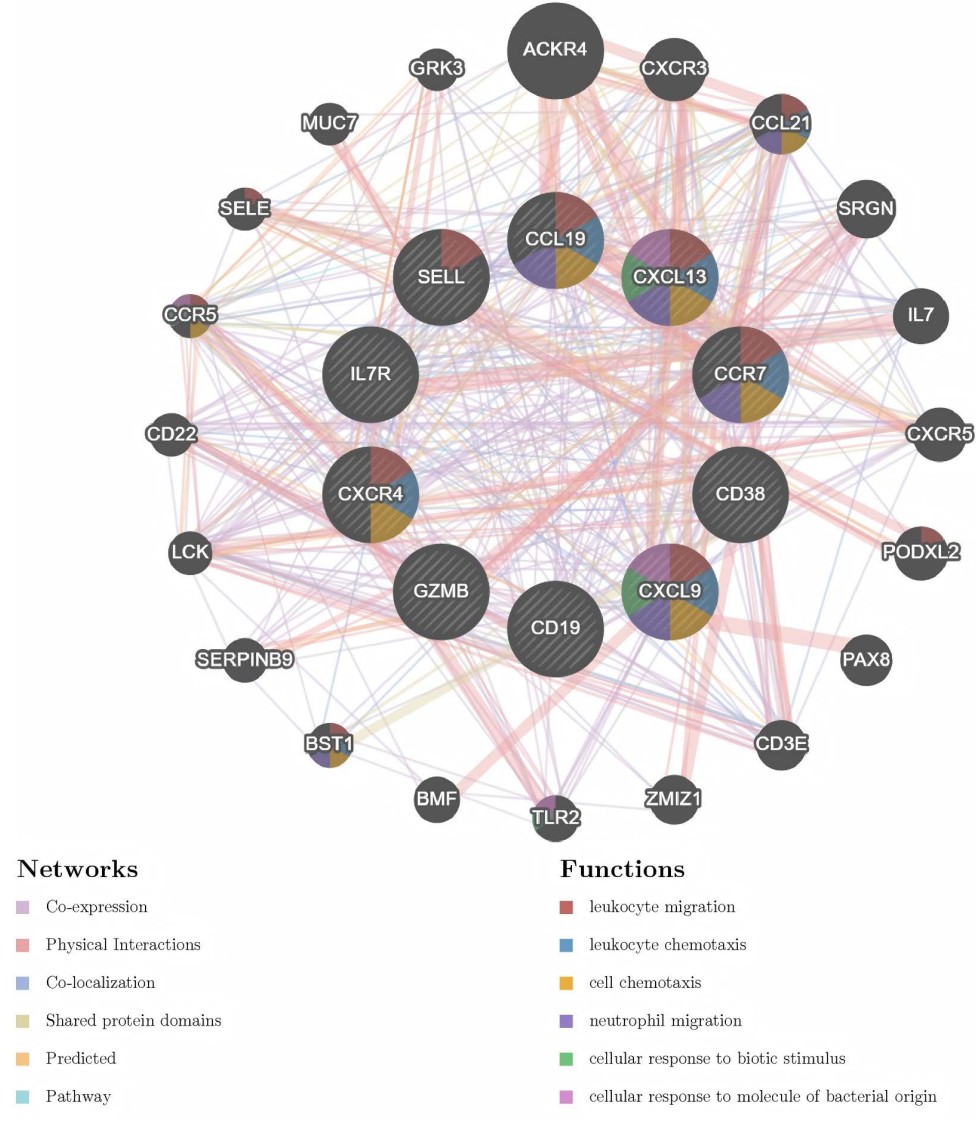

**Networks**

- Co-expression
- Physical Interactions
- Co-localization
- Shared protein domains
- Predicted
- Pathway

**Functions**

- leukocyte migration
- leukocyte chemotaxis
- cell chemotaxis
- neutrophil migration
- cellular response to biotic stimulus
- cellular response to molecule of bacterial origin

**Fig 13. Visualization of the GENEMANIA analysis for commonly expressed differentially expressed genes (DEGs) and their involvement in key biological processes.**

immune infiltration proportions, this significantly enhances their potential as reliable disease biomarkers or therapeutic targets.The complete immune cell expression profiles for the hub genes in rheumatoid arthritis and ulcerative colitis are provided in S13 and S14 Files, respectively (Figs 19 and 20).

**3.8. Subcellular Localization Characteristics of Hub Genes:** Systematic subcellular localization analysis of the five hub genes revealed a distinct compartmentalization pattern: four genes (SELL, CCR7, CD19, and CXCR4, representing 80% of the total) displayed predominant plasma membrane localization. This marked membrane localization preference indicates their encoded proteins primarily function as surface receptors or adhesion molecules. CXCL13 was identified as the sole secreted protein (20%), mainly localized to the extracellular space—a finding fully consistent with its established role as a chemokine and indicative of its involvement in immune microenvironment modulation through paracrine signaling.Reliability

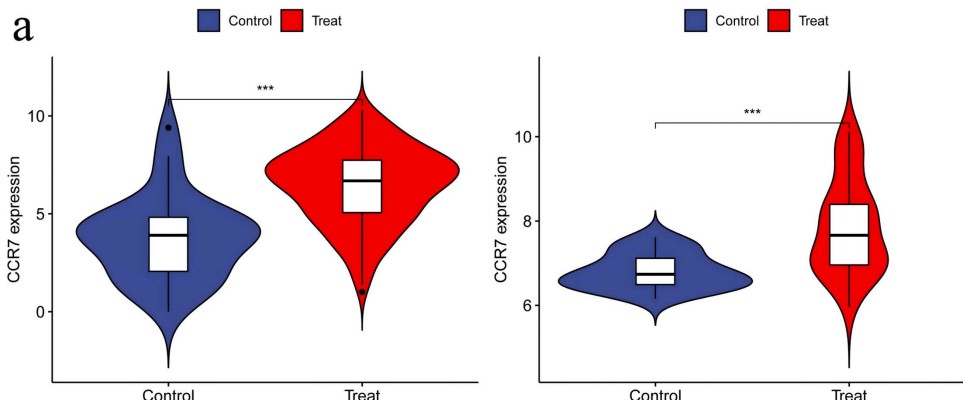

**Fig 14. CCR7 Expression in RA and UC – Demonstrating Significant Upregulation of CCR7 in Both RA and UC Groups, Suggesting Enhanced Activity and Possible Involvement in Disease Processes in the Context of the Disease.**

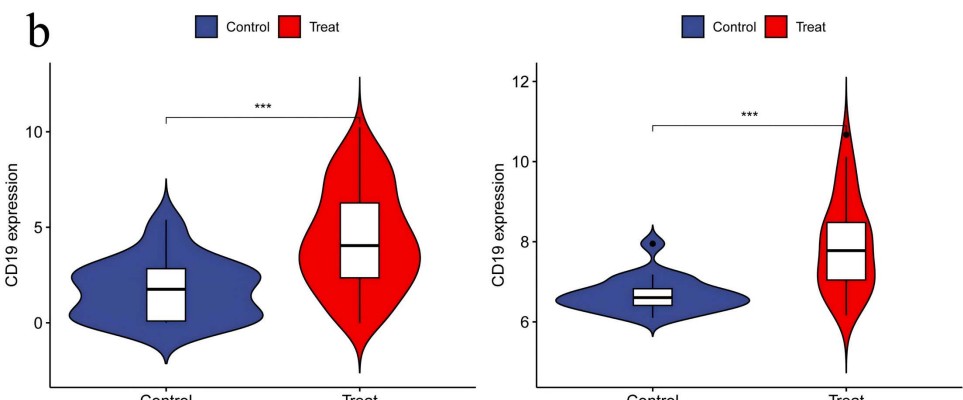

**Fig 15. CD19 Expression in RA and UC – Showing Significant Upregulation of CD19 in Both RA and UC Groups, Indicating Increased Activity and Potential Participation in Disease Processes in the Context of the Disease.**

assessment confirmed high confidence in the localization data (Table 1): 80% of the genes (SELL, CCR7, CXCL13, and CXCR4) were supported by "Enhanced" level evidence, while the remaining gene (CD19) received "Supported" level validation. These high-quality data establish a solid foundation for subsequent mechanistic investigations.

**3.9. Constructing the CeRNA Regulatory Network:** We constructed a ceRNA regulatory network comprising 6 lncRNAs, 4 miRNAs, and 2 core genes, forming a total of 12 regulatory relationships.The complete ceRNA regulatory network is visualized in S1 Fig. Notably, the regulatory axis centered around hsa-miR-622–CXCR4/CCR7 suggests that immune cell migration may represent a shared pathological basis for these two autoimmune diseases. Future studies could experimentally validate the expression levels of key nodes in this network in tissues of RA and UC patients, as well as their impact on inflammatory responses, thereby providing a theoretical foundation for the development of targeted ceRNA therapeutic strategies.

## 4. Discussion

Statistically, the number of patients with rheumatoid arthritis (RA) and ulcerative colitis (UC) worldwide is showing an upward trend [15,16]. Although UC and RA are two distinct diseases, they share certain connections. Both are related to

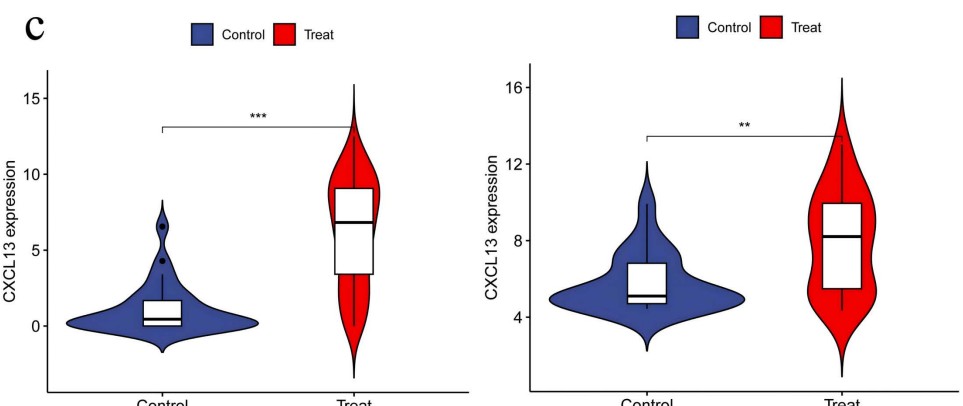

**Fig 16. CXCL13 Expression in RA and UC – Illustrating Significant Upregulation of CXCL13 in Both RA and UC Groups, Reflecting Enhanced Activity and Possible Involvement in Disease Mechanisms in the Context of the Disease.**

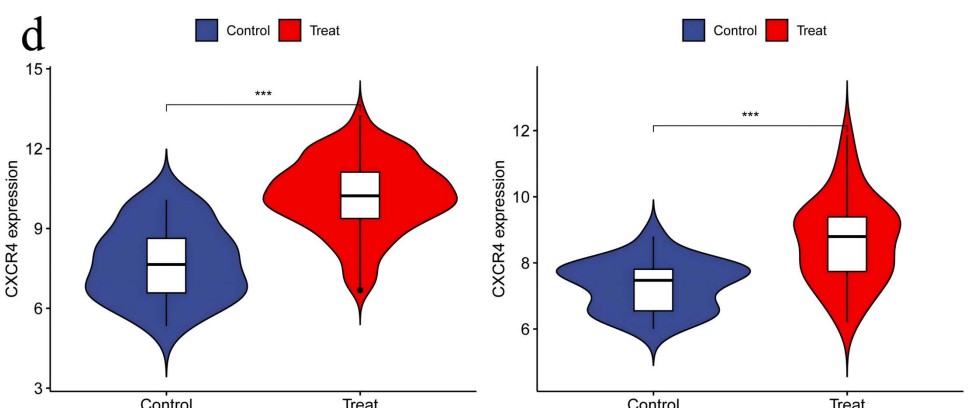

**Fig 17. CXCR4 Expression in RA and UC – Displaying Significant Upregulation of CXCR4 in Both RA and UC Groups, Suggesting Elevated Activity and Potential Role in Disease Processes in the Context of the Disease.**

abnormalities in the immune system, and thus may occur simultaneously in some cases. Studies have shown that patients with UC may have an increased risk of developing RA [17]. This may be because, in some instances, the same immune abnormalities can lead to the onset of both diseases. However, not all UC patients will develop RA, which depends on individual differences and whether timely and effective treatment is received. Recent research progress has revealed overlapping characteristics in the pathogenesis of RA and UC [18]. To investigate these potential links, we employed bioinformatics techniques to analyze the expression patterns of specific genes under both conditions. The results showed higher expression levels of CCR7, CD19, CXCL13, CXCR4, and SELL genes in patients with RA and UC. Our research underscores the pivotal significance of these genes in unraveling the intricate interplay between the shared and distinct pathological processes underlying these autoimmune diseases.

CCR7, as a crucial G-protein-coupled receptor (GPCR), plays a vital dual role in immune responses. It serves as a navigational receptor for leukocyte migration, sensitively perceiving concentration gradients of chemokines and other chemoattractants, and accurately guiding the migration path of individual cells and even cell collectives. More importantly, CCR7 also functions as a builder and regulator of chemotactic gradients, cleverly modulating the spatial and temporal distribution of chemokines by internalizing its specific ligand CCL19, thereby enhancing or prolonging the chemotactic gradient effect.

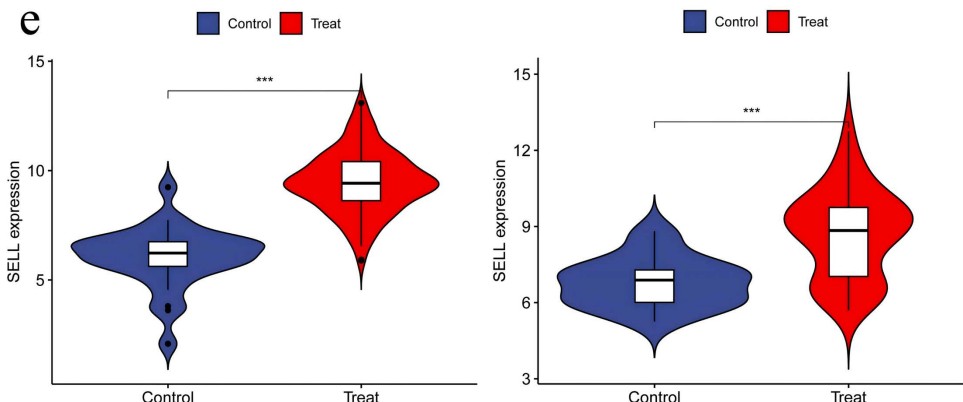

**Fig 18. SELL Expression in RA and UC – Revealing Significant Upregulation of SELL in Both RA and UC Groups, Indicating Increased Activity and Possible Participation in Disease Pathways in the Context of the Disease.**

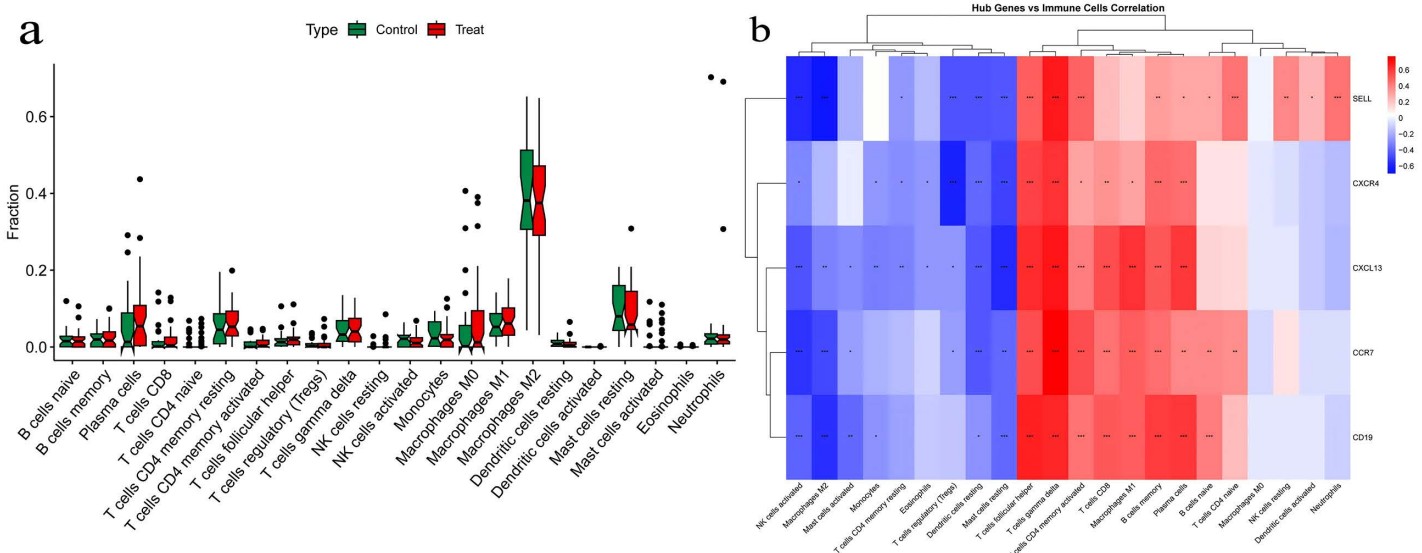

**Fig 19. Immune cell profiling and gene-immune cell correlation in rheumatoid arthritis (RA).** 19a Boxplot of the Infiltration Proportions of 22 Immune Cell Types in Rheumatoid Arthritis 19b Heatmap of Correlation between Hub Genes and Immune Cell Infiltration.

This unique mechanism endows dendritic cells (DCs) with the ability to maintain long-distance navigation and drives the formation of complex and variable collective migration patterns, which can flexibly adapt to the size and shape of different environments while providing essential navigational information for accompanying cells [19]. In Sjögren's syndrome, the expression of CCL19/CCR7 is significantly elevated in salivary gland tissue [20]. Research on multiple sclerosis indicates that cerebrospinal fluid levels of CCL19 correlate with the numbers of T cells and CCR7 + dendritic cells. Furthermore, in experimental autoimmune encephalomyelitis models, CCR7 signaling is involved in immune cell recruitment and the regulation of Th1/Th17 responses [21]. Studies on psoriasis have also reported an association between increased levels of CCR7 + T lymphocytes and disease persistence [22].Studies have shown that in the synovial microenvironment of RA, the maturity of DCs increases, manifested by high expression of CCR7 and other maturation markers, accompanied by a

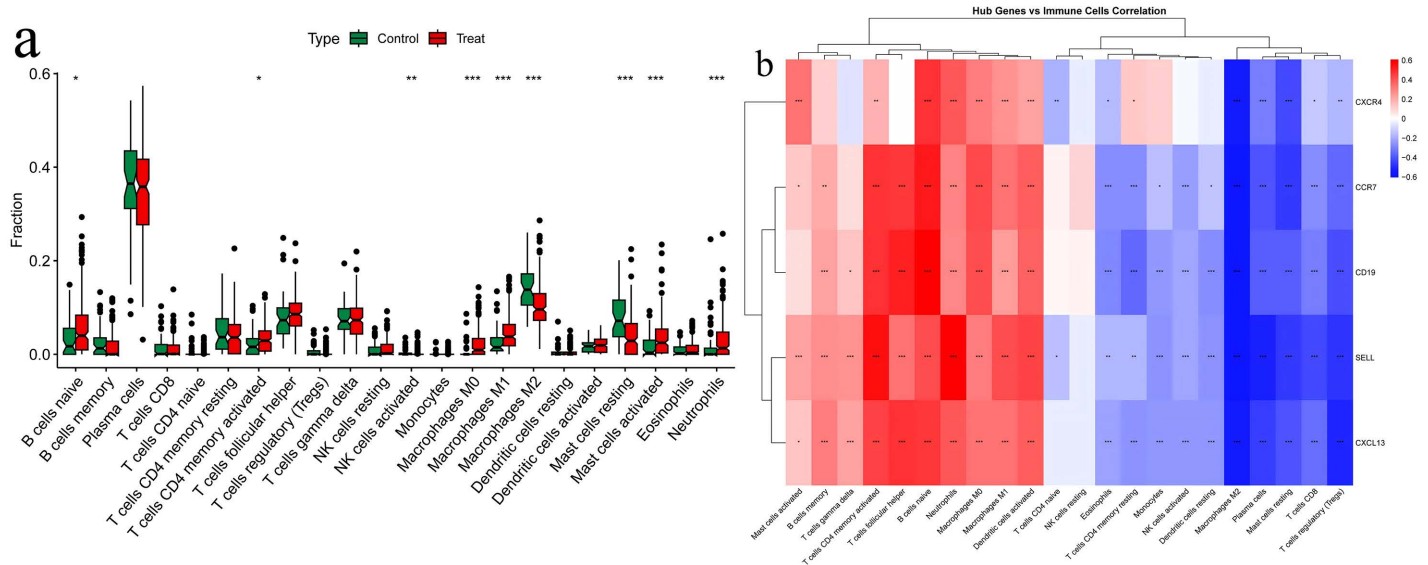

**Fig 20. Immune cell profiling and gene-immune cell correlation in ulcerative colitis (UC).** 20a Boxplot of the Infiltration Proportions of 22 Immune Cell Types in Ulcerative Colitis 20b Heatmap of Correlation between Hub Genes and Immune Cell Infiltration.

**Table 1. Subcellular Localization Analysis of Hub Genes.**

| Gene | Main_Location | Reliability | Location_Category |
|------|---------------|-------------|-------------------|
| SELL | Plasma membrane | Enhanced | Membrane |
| CCR7 | Plasma membrane | Enhanced | Membrane |
| CD19 | Plasma membrane | Supported | Membrane |
| CXCL13 | Secreted | Enhanced | Secreted |
| CXCR4 | Plasma membrane | Enhanced | Membrane |

metabolic shift towards glycolysis [23]. Furthermore, research by Emma Probst Brandum et al. further revealed that CCR7 antagonists may be beneficial in preventing the recruitment of immune cells to inflamed tissues, thereby preventing auto-immune reactions from worsening in the early stages of diseases such as RA, suggesting that in inflammatory diseases like RA, CCR7 may participate in and promote disease progression, and its expression may be relatively high or at least expressed in certain cell types (such as DCs) [24]. For UC, studies have also shown that CCR7 may play an important role. Frank Autschbach et al. tested new reagents on cryosections of normal human intestines, tonsils, and livers, and analyzed inflamed intestinal tissues from patients with Crohn's disease and UC, finding that CCR7 expression was upregulated in inflamed intestinal tissues of UC patients, further suggesting that CCR7 may be involved in the inflammatory process of UC [25]. Jie HE et al. used bioinformatics methods to identify 11 core genes, including CCR7 (such as ICAM1, SELL, CD44, etc.), and pointed out that these genes may play critical regulatory roles in UC, closely related to pathological processes such as lymphoproliferative diseases, inflammation, and necrosis. Subsequent Western Blot experimental results also confirmed the high expression of ICAM1 and SELL in UC, further emphasizing the importance of CCR7 in the pathogenesis of UC [26].The research on CCR7 in the field of immune diseases still holds boundless possibilities. Although numerous studies have currently uncovered the expression changes and potential action mechanisms of CCR7 in various immune diseases, its specific functions in different disease stages and different cell subsets remain to be further explored in depth.

CD19 is a CD molecule (i.e., leukocyte differentiation antigen) expressed on B cells, belonging to the Ig superfamily. It participates in the interaction between B cells and other immune cells such as T cells and macrophages, regulating humoral and cellular immune responses. It also modulates the transport of $Ca^{2+}$ within B cells. CD19 influences the activation and proliferation of B cells, thereby regulating the intensity and duration of immune responses [27]. Research by Anang DC et al. revealed a notable phenomenon: compared to lymphoid tissues from healthy individuals, the number of CD19+B cells, CD4+CXCR5+follicular helper T cells, and CD8+CXCR5+follicular T cells is significantly increased in the lymphoid tissues of patients with rheumatoid arthritis (RA) and individuals at risk for RA [28]. This discovery further emphasizes the potential importance of CD19 in the pathogenesis of RA. Relevant research by Heung Bum Lee, MD indicates that the percentage of CD19 is elevated in patients with ulcerative colitis (UC) compared to a control group [29]. In addition to RA and UC, CD19 has also garnered attention in other autoimmune diseases. In patients with systemic lupus erythematosus (SLE), a significantly higher frequency of CD19+CD20−B cells is observed compared to healthy controls, and this frequency positively correlates with disease activity [30].In multiple sclerosis (MS), CD19+B cells participate in inflammatory responses within the central nervous system, promoting the development of neuroinflammation through interactions with T cells and other immune cells [31].These studies collectively indicate that CD19 plays a pivotal role in various autoimmune diseases.However, to more comprehensively and accurately reveal the expression characteristics and potential action mechanisms of CD19 in a wider range of immune diseases such as ulcerative colitis (UC) and rheumatoid arthritis (RA), it is still necessary to conduct larger-scale and more rigorously designed scientific studies for in-depth exploration and validation.

CXCL13 is described as the most effective chemokine for B cells [32].Through interaction with its receptor CXCR5, it effectively promotes the migration of B lymphocytes. The binding of CXCL13 to CXCR5 also contributes to the maturation of B cells into plasma cells, which then produce antibodies [33]. CXCL13 regulates the activation and migration of various immune cells at inflammatory sites, mediating the expression of inflammatory mediators and thereby modulating inflammatory responses,It plays a crucial role in autoimmune diseases such as Sjögren's syndrome, lupus nephritis, and multiple sclerosis [34,35]. Achudhan D et al. found that levels of CXCL13 and TNF-α were higher in Rheumatoid Arthritis (RA) samples compared to healthy controls [36]. Another meta-analysis involving 332 RA patients and 147 healthy controls also pointed out that circulating CXCL13 levels in RA patients were significantly higher than those in healthy individuals, further confirming the crucial role of CXCL13 in the pathogenesis of RA [37].Udai P Singh et al. compared the systemic concentrations of key chemokines and cytokines in 42 patients with inflammatory bowel disease (IBD) of different disease activities with the levels in 10 healthy donors. They found that, compared to normal healthy donors, a series of chemokine levels were significantly increased in IBD patients, including macrophage migration inhibitory factor (MIF), CCL25, CCL23, CXCL5, CXCL13, CXCL10, CXCL11, MCP1, and CCL21 ($P < 0.05$) [38]. Research by Lu Hui et al..also demonstrated that CXCL13 is significantly overexpressed in both ulcerative colitis (UC) and rheumatoid arthritis (RA), closely related to disease activity and the severity of inflammatory responses. In UC, CXCL13 is mainly secreted by T peripheral helper cell cells and macrophages; in RA, CXCL13 is mainly secreted by various cells in synovial tissue, including monocytes/macrophages, T follicular helper cells, and Follicular dendritic cells [39].

CXCR4 is an amino acid rhodopsin-like G protein-coupled receptor (GPCR) that specifically binds to the ligand CXCL12. CXCL12 exhibits a strong chemotactic effect on lymphocytes, and as the exclusive receptor for CXCL12, CXCR4 can activate multiple signaling pathways upon activation by CXCL12, thereby regulating cell migration, survival, and proliferation [40] The interction between such receptors and ligands holds profound significance in both biological and medical fields. Numerous studies have demonstrated that the CXCR4/CXCL12 axis exhibits abnormal activation in a variety of autoimmune diseases, including psoriasis, systemic lupus erythematosus (SLE), multiple sclerosis (MS), rheumatoid arthritis (RA), type 1 diabetes (T1D), and inflammatory bowel disease (IBD), particularly ulcerative colitis (UC). For instance, elevated mRNA expression of CXCR4/CXCL12 has been observed in both psoriatic lesions and SLE patients [41,42]. Focusing on rheumatoid arthritis, which is the focus of our current analysis, the pathogenic role of

CXCR4 is particularly prominent. In a prospective study by I B Hansen et al. on rheumatoid arthritis (RA) patients receiving methotrexate (MTX) treatment, plasma CXCL12 (p-CXCL12) levels in RA patients were significantly and persistently elevated compared to the control group [43]. Additionally, other studies have shown that the expression of CCR1, CCR2, CCR4, CCR5, and CXCR4 on the surface of B cells in synovial fluid (SF) of arthritis patients is significantly increased [44]. In studies of ulcerative colitis (UC), CXCR4 expression also showed a significant increase. Research by S Hosomi et al. found that the number of immature plasma cells in the peripheral blood of patients with active ulcerative colitis was significantly increased, and these cells expressed positive for multiple chemokine receptors, including CXCR4. Compared with the healthy control group, CXCR4 expression levels in UC patients were significantly elevated. Further research suggested that high CXCR4 expression may be closely related to the migration of immature plasma cells to the inflammatory sites in UC, revealing the important role of CXCR4 in the pathogenesis of ulcerative colitis [45]. In summary, the CXCR4/CXCL12 axis represents a shared immune dysregulation pathway connecting multiple autoimmune diseases.

SELL (selectin L) is a membrane glycoprotein widely expressed on the surface of human and other animal cells, primarily expressed on the surface of endothelial cells and upregulated during inflammation and immune responses, mediating the migration and infiltration of leukocytes. It plays a crucial role in the initial "rolling" adhesion of leukocytes on the vascular endothelial surface, enabling immune cells to scan chemotactic signals and select appropriate tissue entry sites [46,47], It exhibits abnormal phenotypes in a variety of autoimmune diseases, is closely associated with disease activity and immune cell migration, and influences the onset and progression of these diseases. Research by Y. Kurohori MD et al. on peripheral blood mononuclear cells (PBMCs) from rheumatoid arthritis (RA) patients revealed that the positive rate of L-selectin in RA patients was significantly higher than that in the normal population. Furthermore, in active RA cases, the expression of L-selectin-positive CD4 + cells and the L-selectin/CD4 ratio was also significantly higher than in inactive cases or the normal control group. More interestingly, the number of L-selectin-positive cells showed a positive correlation with laboratory indicators such as the erythrocyte sedimentation rate (ESR) and C-reactive protein (CRP), further strengthening the close link between L-selectin expression and RA inflammation activity [48]. Similarly, elevated SELL mRNA expression has also been observed in systemic sclerosis [49], and upregulation of SELL has been detected in ankylosing spondylitis as well [50]. Although significant progress has been made in research on SELL (L-selectin) in autoimmune diseases, many aspects of its underlying mechanisms remain unclear. In the future, large-scale, multicenter clinical studies are needed to further explore the specific molecular mechanisms regulating SELL expression.

Previous studies have clearly indicated that microRNAs (miRNAs) and their related genes occupy a central position in the regulatory network of inflammatory bowel disease (IBD). Specifically, research by Mohsen Nemati Bajestan et al. revealed that in the IBD environment, both the long non-coding RNA MALAT1 and the pro-inflammatory cytokine IL-6 expression are significantly upregulated, and their interaction is finely regulated by microRNAs, including hsa-miR-9-5p (abbreviated as miR-9-5p). This finding strongly supports the presence and indispensable regulatory role of miR-9-5p in the pathophysiology of IBD [51]. Furthermore, our research explored the potential connection between rheumatoid arthritis (RA) and ulcerative colitis (UC) at the miRNA and gene levels, particularly focusing on the two key genes CCR7 and CXCR4. However, this preliminary finding requires extensive subsequent research for further validation and expansion to more comprehensively reveal its underlying mechanisms and clinical significance.

In general, our exploration of the molecular mechanisms of rheumatoid arthritis (RA) and ulcerative colitis (UC) has revealed significant similarities between these two diseases. This provides new insights into their pathological mechanisms and valuable perspectives for the development of new treatment strategies and the identification of potential therapeutic targets for RA and UC. However, current research is still primarily based on bioinformatics analysis, without in vitro functional experiments or animal model validation. Therefore, further studies are required to clarify the specific roles of these genes in disease progression and to evaluate their feasibility as therapeutic targets or diagnostic biomarkers.

## 5. Conclusion

This study employed bioinformatics approaches to conduct an in-depth analysis of gene expression differences between two autoimmune diseases RA and UC. Based on differential expression analysis, we identified genes differentially expressed in each disease compared to normal controls and further isolated a shared core set of disease-related genes. The results reveal that although RA and UC differ in clinical manifestations and affected organs, they exhibit significant overlap in certain immune-related signaling pathways and gene expression patterns, suggesting common underlying mechanisms in immune regulation and inflammatory responses. Notably, several key genes involved in immune cell activation and inflammatory mediator release showed consistent dysregulation in both diseases, highlighting them as priority candidates for subsequent functional experiments and translational research. In summary, this study elucidates both shared and distinct gene expression features between RA and UC from a bioinformatics perspective, providing new insights into their common and specific pathological mechanisms. The identified common key genes establish a theoretical foundation for exploring cross-disease immunomodulatory targets, holding significant scientific and potential translational value.

## 6. Limitations

While this study systematically identified shared hub genes and pathways between rheumatoid arthritis (RA) and ulcerative colitis (UC) through bioinformatic analysis, several limitations should be acknowledged. First, our analysis relied on bulk tissue sequencing data, the results of which are inevitably influenced by tissue cellular heterogeneity. Inter-individual variations in the proportions of immune cells, stromal cells, and other components within synovial tissue and intestinal mucosa may represent significant confounding factors in gene expression variation. Although we employed bioinformatic methods for estimation and statistical adjustment, residual confounding effects on the accuracy of the results cannot be entirely excluded.Second, the integration of datasets was challenged by incomplete clinical metadata. The absence of detailed information such as age, sex, disease stage, medication history, and comorbidities in some datasets limited our ability to more comprehensively control for these potential confounding variables in the models, which may affect the specificity of the differential expression analysis.Furthermore, a primary limitation of this study is that all findings are derived from bioinformatic inferences and have not yet been validated through functional experiments. The causality of the identified hub genes and their predicted molecular mechanisms, along with their precise biological functions in the pathogenesis of RA and UC, require further confirmation using in vitro cell models, organoids, or animal studies.Finally, the retrospective nature of analyses based on public databases introduces an inherent risk of selection bias. Additionally, technical variations across different study platforms, despite normalization, may retain residual batch effects.

## Supporting information

**S1 Table. Details of rheumatoid arthritis-related datasets used in this study.**
(DOCX)

**S2 Table. Details of the ulcerative colitis datasets used in this study.**
(DOCX)

**S1 File. Rheumatoid Arthritis dataset before normalization.**
(CSV)

**S2 File. Rheumatoid Arthritis dataset after.**
(CSV)

**S3 File. Ulcerative Colitis dataset before normalization.normalization.**
(CSV)

**S4 File. Ulcerative Colitis dataset after normalization.**
(CSV)

**S5 File. Differentially expressed genes for rheumatoid arthritis.**
(CSV)

**S6 File. Differentially expressed genes for ulcerative colitis.**
(CSV)

**S7 File. Hub genes from PPI network (confidence threshold = 0.300).**
(CSV)

**S8 File. Hub genes from PPI network (confidence threshold = 0.400).**
(CSV)

**S9 File. Hub genes from PPI network (confidence threshold = 0.500).**
(CSV)

**S10 File. Intersection of hub genes from thresholds 0.3, 0.4, and 0.5.**
(CSV)

**S11 File. ROC analysis results for hub genes in rheumatoid arthritis.**
(CSV)

**S12 File. ROC analysis results for hub genes in ulcerative colitis.**
(CSV)

**S13 File. Immune cell expression profiles of hub genes in rheumatoid arthritis.**
(CSV)

**S14 File. Immune cell expression profiles of hub genes in ulcerative colitis.**
(CSV)

**S1 Fig. ceRNA Regulatory Network.**
(TIF)

**S1 Data.**
(ZIP)

## Author contributions

**Conceptualization:** Peng-fei han, Tao Wu.

**Data curation:** Wei-rong Cui, Fang-zheng He, Tao Wu.

**Investigation:** Fang-zheng He.

**Supervision:** Peng-fei han.

**Writing – original draft:** Wei-rong Cui, Chang-Sheng Liao.

**Writing – review & editing:** Peng-fei han, Chang-Sheng Liao.

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
