## [Decision Letter · Decision Letter 0]

3 Jun 2025

Dear Dr. Han,

Thank you for submitting your manuscript to PLOS ONE. After careful consideration, we feel that it has merit but does not fully meet PLOS ONE’s publication criteria as it currently stands. Therefore, we invite you to submit a revised version of the manuscript that addresses the points raised during the review process.

We have received the expert reviewer's opinions and invite you to submit a revised manuscript version. Please consider and address each of the comments raised by the reviewers.  

We look forward to receiving your revised manuscript.

Kind regards,

Senthilnathan Palaniyandi, Ph.D

Academic Editor

PLOS ONE

Journal Requirements:

3. Thank you for uploading your study's underlying data set. Unfortunately, the repository you have noted in your Data Availability statement does not qualify as an acceptable data repository according to PLOS's standards.

Reviewers' comments:

Reviewer's Responses to Questions

**Comments to the Author**

1. Is the manuscript technically sound, and do the data support the conclusions?

Reviewer #1: Yes

Reviewer #2: Partly

2. Has the statistical analysis been performed appropriately and rigorously?

Reviewer #1: Yes

Reviewer #2: Yes

3. Have the authors made all data underlying the findings in their manuscript fully available?

Reviewer #1: Yes

Reviewer #2: Yes

4. Is the manuscript presented in an intelligible fashion and written in standard English?

Reviewer #1: Yes

Reviewer #2: No

Reviewer #1: This manuscript presents a bioinformatics-based comparative analysis of gene expression profiles in rheumatoid arthritis (RA) and ulcerative colitis (UC) using publicly available microarray datasets. The goal of identifying shared and disease-specific molecular signatures is clinically relevant and potentially impactful. The authors have utilized standard tools and multiple datasets, which strengthens the analytical rigor. However, the manuscript requires some revisions to improve clarity and the biological interpretation of results.

1. Overstatement of Therapeutic Implications: The manuscript repeatedly refers to the identified genes as therapeutic targets or biomarkers. While these genes may warrant further investigation, the current study is based solely on in silico transcriptomic analysis without functional validation. Please moderate the language to reflect that these genes are potential candidates for future research, not confirmed therapeutic targets as the manuscript addresses very common genes (e.g. CD19 on B cells) that needs further proof to be considered a specific biomarker or therapeutic target for these diseases.

2. Lack of RA vs. UC Differential Comparison: Despite the stated objective of uncovering molecular mechanisms that distinguish RA from UC, the manuscript focuses primarily on genes commonly upregulated in both diseases. Please modify the objective of the manuscript to reflect better the data evaluated.

3. Biological Interpretation of Results: In the section “GENEMANIA Online Analysis” authors list the ten hub genes and suggest that these genes are involves in metabolic and thermogenic processes. Further down in the discussion they go in detail explaining the role of each gene where they correctly explain the functions of these genes – nonrelated to metabolic and thermogenic processes but strong inflammatory responses and cellular migration. The authors should revise this interpretation or provide robust references to support the claim.

4. Review grammar, typographical errors and abbreviations: Please review the manuscript for grammar errors, long sentences or typographical errors to improve readability. In the results section match UC abbreviation with the rest of the manuscript. Add the full word phrase for Tph, WB and other abbreviations lacking explanation.

5. Attach better figure quality.

Reviewer #2: The study reports the Common Differential Gene Expression between Rheumatoid Arthritis (RA) and Ulcerative Colitis (UC) using datasets through bioinformatics analysis. However, there are key concerns which need to be addressed.

• The rationale for choosing these disease conditions RA and UC is not clear. Why did they did not choose other autoimmune conditions which are closely related to UC or RA.

• The reason for choosing these specific datasets is not clearly mentioned.

• Did they look at the other parameters like gender, age, treatment conditions and other existing co-morbidities which could influence the study outcomes.

• The results report the differentially expressed genes (DEG’s) such as CD19, CCR7 which are very common in other autoimmune conditions. The relevance of the DEG’s with other autoimmune disorders should be discussed in detail.

• The identified DEG’s should be validated experimentally to address the common molecular mechanisms.

• The quality of the figures should be improved, and the study can be strengthened by providing the detailed insights into relationship between these conditions.

**Do you want your identity to be public for this peer review?** For information about this choice, including consent withdrawal, please see our Privacy Policy

Reviewer #1: No

Reviewer #2: No

---

## [Author Response · Author response to Decision Letter 1]

25 Jul 2025

Dear Reviewers,

We wholeheartedly extend our gratitude to all of you for the precious feedback on our manuscript titled "Analysis of Common Differential Gene Expression between Rheumatoid Arthritis and Ulcerative Colitis". We are well aware of the immense effort and time you have devoted during the review process, and we hereby offer our sincere thanks. Your insightful remarks have not only enhanced our in-depth understanding of this research but also provided us with invaluable suggestions. We have carefully contemplated all the suggestions and will incorporate them one by one in the revised manuscript. Below are our point-by-point responses to each of your major comments.

Reviewer #1: 

1.Dear Reviewer, I sincerely appreciate your invaluable review comments and your attention to enhancing the quality of this manuscript. Your observation regarding the "overstatement of therapeutic significance" is exceptionally insightful and of paramount importance. Indeed, it is scientifically rigorous to refer to the identified genes as "therapeutic targets" or "biomarkers" without functional validation, particularly for widely expressed genes like CD19. Following your suggestion, we have revised the relevant sections of the manuscript to present the potential research value of these genes in a more cautious manner.Terminology Revision: We have replaced the term "therapeutic targets" with the more prudent phrase "potential candidate genes for future research," emphasizing that these genes require further functional validation and clinical evidence before they can be confirmed as genuine therapeutic targets or biomarkers. Additionally, we have refined the language in certain paragraphs to prevent overinterpretation or exaggeration of the current study's conclusions.

2.Dear Reviewer,Thank you for reviewing my submission and providing your invaluable feedback. The issue you raised regarding the "lack of comparative analysis between rheumatoid arthritis and ulcerative colitis" is of critical importance. Although the primary objective of this study was to identify potential shared molecular mechanisms between RA and UC , as you correctly pointed out, the current research primarily focused on genes that were concurrently upregulated in both diseases, without systematically comparing their differential characteristics in terms of gene expression patterns, functional pathways, or pathological mechanisms. To better reflect the scientific value of this work, we have revised the research objectives to more precisely define the scope and focus. Specifically, the updated objectives now state:This study employs bioinformatics analysis with the objective of identifying commonly differentially expressed genes (DEGs) in ulcerative colitis (UC) and rheumatoid arthritis (RA), as well as exploring their underlying molecular mechanisms. By doing so, it aims to provide a theoretical basis for investigating the potential associations between these two diseases and developing novel therapeutic strategies.

3.Dear Reviewer,Thank you for reviewing my submission and providing your valuable feedback. In the section on "GENEMANIA online analysis," we listed ten hub genes and initially suggested that they might be involved in metabolic and thermogenic processes. However, later in the discussion section, we provided a more detailed description of these genes' functions, highlighting their primary associations with inflammatory responses and cell migration, rather than metabolic or thermogenic processes. We sincerely apologize for this inconsistency in our statements. To ensure the scientific rigor and logical coherence of the paper, we have revised the relevant sections. Specifically, we have re-examined and adjusted the descriptions in the "GENEMANIA online analysis" section to more accurately reflect the functional associations of these genes within the gene network. Furthermore, in the discussion section, we have further clarified the biological functions of these genes. For instance, we have removed the statement "involved in metabolic and thermogenic processes" and supplemented it with a more precise description: their central roles in various immune and inflammation-related biological processes, as well as their participation in immune cell activation, recruitment, and migration.

4.Dear Reviewer, I'm deeply grateful for your meticulous review of my submission and the precious feedback you've offered. We've conducted a systematic revision targeting language and structural issues: First off, we've gone through the manuscript sentence by sentence to rectify all spelling and grammatical errors. Next, we've disassembled and reconstructed lengthy and complex sentences, and employed logical connectors to enhance the coherence between paragraphs. Especially in the Methodology and Results sections, we've sharpened the content by trimming repetitive descriptions and consolidating similar information to boost its conciseness.

At the same time, we've re-examined and reorganized the scientific logical chain of the study. While keeping the core data intact, we've eliminated redundant expressions and supplemented key research details to enrich the completeness of the information.

In the Results section, we've annotated the full names for all abbreviations when they first appear in each paragraph. For the same abbreviations that show up later in the paragraph, we've uniformly used lowercase letters. Regarding unexplained abbreviations in the original manuscript, such as "Tph" and "WB," we've added their corresponding full phrases to ensure that readers can clearly comprehend the content.

5.Dear Reviewer,We have conducted a thorough inspection and optimization of the relevant figures and tables. We have adjusted their sizes and resolutions in accordance with the journal's requirements to ensure they meet the submission standards while maintaining both clarity and aesthetic appeal.

Reviewer #2: •

1.Dear Reviewer,Thank you for raising these significant questions. Regarding the selection of rheumatoid arthritis (RA) and ulcerative colitis (UC) as research subjects, we have made revisions in the manuscript based on the following scientific considerations:

(1) From an epidemiological perspective, both RA and UC have relatively high incidence rates globally, with a rising prevalence in recent years. Moreover, these two diseases exhibit a considerable overlap rate in clinical practice, particularly among UC patients, who have a significantly higher incidence of extraintestinal manifestations (such as arthritis) compared to patients with other intestinal diseases.

(2) In terms of pathological mechanisms, both RA and UC are chronic inflammatory diseases. RA primarily involves chronic inflammation of the joint synovium, while UC focuses on the inflammatory response in the intestinal mucosa. Despite their different lesion sites, both diseases are characterized by abnormal activation of the immune system and chronic inflammatory responses (e.g., Fcγ receptor signaling). Additionally, there is notable genetic overlap between RA and UC in the IL-23/Th17 pathway, providing an ideal comparative model for studying immune-related mechanisms.

(3) Research value and innovation: Selecting RA and UC as research subjects not only helps to uncover the common mechanisms underlying autoimmune diseases but also provides a theoretical basis for developing new therapeutic strategies. We have supplemented the research background section to more clearly explain the reasons for choosing RA and UC and emphasize their representativeness and scientific value in autoimmune disease research.

We highly value your revision suggestions and have made corresponding adjustments to the manuscript according to your feedback.

2.Dear Reviewer, The gene expression datasets we selected are all sourced from the public bioinformatics database—GEO (Gene Expression Omnibus). GEO is a high-throughput gene expression data repository maintained by the National Center for Biotechnology Information (NCBI) in the United States. It encompasses a wide range of research fields and features high-quality data with a broad clinical application background. The datasets we chose include samples from patients with rheumatoid arthritis (RA) and ulcerative colitis (UC), as well as healthy control groups, with each group comprising more than 20 samples. This sample size ensures the statistical significance of our experimental results and helps avoid phenotypic confounding due to insufficient sample sizes. Additionally, we prioritized datasets that align with our specific research objectives to ensure data comparability and the reliability of our analyses.

3.Dear Reviewer, Thank you for your questions. We hereby provide the following responses to the issues you raised: We explicitly acknowledge that parameters such as age, gender, treatment conditions, and comorbidities may potentially confound gene expression profiles and serve as important background variables in certain analyses. However, due to limitations in the data collection process, we were unable to obtain detailed age group and gender information for all samples. Given that our primary research objective is to minimize batch effects across datasets through rigorous normalization procedures, identify differentially expressed genes associated with rheumatoid arthritis and ulcerative colitis, and explore the roles of these genes in disease progression, we decided to include the aforementioned datasets in our study and performed standardization processes during subsequent data analysis. We believe that this approach can, to a certain extent, balance the influence of confounding factors while focusing on the core objectives of our research. This aspect has been clarified in the "Limitations" section of our article.

4.Dear Reviewer,Thank you for this valuable suggestion. Delving into the performance of these differentially expressed genes (DEGs) in other autoimmune diseases will indeed facilitate a better understanding of the potential significance of our findings. In response to your advice, we have added supplementary discussions in the relevant section. We believe that these additional analyses have substantially enhanced the depth and breadth of our study, enabling readers to gain a more comprehensive understanding of the disease-specific and shared characteristics of these DEGs.

5.Dear Reviewer,We sincerely appreciate your valuable feedback. We have come to deeply recognize the pivotal role that experimental validation plays in research. In this study, we conducted preliminary validation using an additional dataset. Given the large number of genes initially screened, validating each one individually would lack focus and efficiency. Therefore, we plan to further screen for 1 - 3 ideal genes through the construction of animal models in the later stage for subsequent validation. However, constructing and validating animal models takes time. Thus, we intend to report the above experimental content and validation results in a forthcoming paper.

The current research findings serve as the preliminary results for subsequent experiments. Nevertheless, it is undeniable that they lack the support of replicable experimental results. Hence, we have added a detailed explanation of this limitation and our future prospects in the "Limitations" section of the manuscript.Once again, we thank you for your precious and insightful suggestions, which have been of immense help in improving our research and enhancing its scientific quality.

6.Dear Reviewer,Regarding the issue of the images, we have utilized the official PLOS ONE processing platform PACE (https://pacev2.apexcovantage.com/ Home Page - Apex.PACE ) to adjust the dimensions and resolution of the charts and graphs. This ensures that they meet the submission requirements while maintaining both clarity and aesthetic appeal.As for the detailed explanation you requested concerning the relationship between rheumatoid arthritis (RA) and ulcerative colitis (UC), we have made supplementary enhancements in the revised manuscript, primarily based on the following considerations: We explored the connections between the extra-articular manifestations of RA and UC. In terms of immune pathways, both diseases are characterized by abnormal activation of the immune system and chronic inflammatory responses. Additionally, there is a significant genetic overlap in the IL-23/Th17 pathway between the two. Moreover, they share common genetic and molecular mechanisms. We delved into the genetic overlaps between RA and UC, such as the differential expression of genes like CCR7, CD19, CXCL13, CXCR4, and SELL, and their roles in the diseases.However, it is important to note that current research still requires stronger experimental validation to confirm the practical clinical value of the identified genes and pathways.

We again thank the reviewer for their valuable comments and look forward to your further feedback on our revised manuscript.

Best regards,

Wei-rong Cui

---

## [Decision Letter · Decision Letter 1]

5 Oct 2025

Dear Dr. %Peng-fei han%,

Thank you for submitting your manuscript to PLOS ONE. After careful consideration, we feel that it has merit but does not fully meet PLOS ONE’s publication criteria as it currently stands. Therefore, we invite you to submit a revised version of the manuscript that addresses the points raised during the review process.

We look forward to receiving your revised manuscript.

Kind regards,

Srinivas Mummidi, D.V.M., Ph.D.

Academic Editor

PLOS ONE

Journal Requirements:

Additional Editor Comments (if provided):

1. A major concern of this paper is using mixed platforms (especially in the validation studies. This could result in false positives. The authors should indicate how they have controlled for this in the discussion

2. GSE92415 is from a golimumab trial and contains pre/post treatment samples; the authors should explicitly state which samples are used in their analysis in such datasets as drugs will have strong effects on gene expression and could bias their results

3. The pipeline description is ambiguous and mixes RNA seq and microarray methods (DESeq2 vs limma) and statistical tests (limma’s moderated t vs a standard two sample t), reducing reproducibility and potentially invalidating some results. Please clarify in Methods how this was done. Ideally Limma should be done for microarray studies and DE-Seq2 for RNA-Seq studies

4. Some statistical analysis descriptions are confusing. Ideally all microarray analyses should have been with a microarray appropriate pipeline (RMA/quantile normalization → probe to gene mapping → limma), and reserve DESeq2 exclusively for RNA seq (GSE89408). Clearly separate these in the Methods. For validation analyses, avoid plain two sample t tests across thousands of genes; either use limma again (with FDR control), or pre specify a small gene set tested with a priori hypotheses.

5. Batch correction is showed in PCAs. Also show the PCAs by case/control to show correction as COMBAT can potentially remove true biological signal. Perform cell composition adjustment and stratified analyses -- otherwise the results are confounded

6. Cytoscape’s cytoHubbacytoHubba MCC refers to Maximal Clique Centrality and not “modularity class.” Could the authors please clarify.

7. The hub list is very generic. The robustness of the hub ranking could be tested using network evidence type edge confidence thresholds, and network randomization, and report stability. Interpret hubs in the context of cellular compartment. 3. Address tissue heterogeneity and immune cell confounding.

8. For each gene, report log2FC with 95% CI, BH adjusted p in validation, and ROC/AUC for disease vs control. Perform study wise replication (each dataset left out in turn).

9. Provide a fully reproducible ceRNA pipeline (databases, versions, cutoffs, intersection logic) and test for co expression consistency across cohorts; otherwise move the ceRNA figure to Supplementary as hypothesis generating

10. The Discussion section sometimes overstates clinical translation (e.g., “potential therapeutic targets”) on the basis of cross sectional gene expression and network centrality. This not causality -- revise appropriately.

11. Please deposit: (i) the exact R scripts/notebooks, (ii) the sample inclusion lists per GEO series (with platform IDs and tissue source), (iii) the ComBat corrected expression matrices (with and without covariate design), and (iv) full DEG tables

Reviewers' comments:

Reviewer's Responses to Questions

**Comments to the Author**

Reviewer #1: All comments have been addressed

2. Is the manuscript technically sound, and do the data support the conclusions?

Reviewer #1: Yes

3. Has the statistical analysis been performed appropriately and rigorously?

Reviewer #1: Yes

4. Have the authors made all data underlying the findings in their manuscript fully available?

Reviewer #1: Yes

5. Is the manuscript presented in an intelligible fashion and written in standard English?

Reviewer #1: Yes

Reviewer #1: Thank you for addressing the comments. I believe the edits have improved the quality of the manuscript.

**Do you want your identity to be public for this peer review?** For information about this choice, including consent withdrawal, please see our Privacy Policy

Reviewer #1: No

---

## [Author Response · Author response to Decision Letter 2]

23 Nov 2025

Dear Reviewers,

We sincerely appreciate the valuable comments and suggestions regarding our manuscript entitled "Differential Gene Expression Analysis of Common Signatures Between Rheumatoid Arthritis and Ulcerative Colitis." We are truly grateful for the considerable time and effort you have dedicated to the review process. Your insightful critiques have not only deepened our understanding of this study but have also provided us with invaluable recommendations. We have carefully considered each comment and will address them point by point in the revised version. Below are our detailed responses to your primary concerns.

1.We sincerely thank the reviewer for this valuable comment. The reviewer rightly pointed out that integrating datasets from different platforms in validation studies introduces potential batch effects as a critical issue, which may introduce technical biases and lead to false positive findings. We fully agree with this perspective and have prioritized this consideration in our analytical process. Prior to the integrative analysis of validation datasets from different platforms, we specifically applied the ComBat algorithm from the R package "sva" to correct for batch effects in the gene expression matrix. This method is widely recognized as a gold standard for addressing batch effects in genomic data, effectively identifying and removing non-biological variations introduced by technical differences across experimental platforms. Following batch effect correction, we performed differential expression analysis using the R package "limma". Limma accounts for various sources of variation within linear models, thereby yielding more accurate statistical inferences. For the validation phase, our approach primarily involved conducting differential expression analysis within each independent validation dataset to assess whether our core findings could be consistently replicated across different platforms and cohorts. This cross-platform reproducibility substantially reduces the likelihood of false positives and strongly supports the robustness of our conclusions.

2.We sincerely thank you for raising this important and insightful point. You correctly noted that the GSE92415 dataset, derived from a golimumab clinical trial, includes paired samples collected both before and after treatment, and that the pharmacological intervention could significantly alter gene expression profiles. We fully agree that carefully accounting for treatment status is crucial in the analytical design.Upon careful re-examination, we confirm that our initial analysis did not adequately distinguish between pre- and post-treatment samples within the GSE92415 dataset, instead analyzing all samples collectively, which may have introduced confounding bias due to drug effects. We have promptly reanalyzed the data, utilizing only the pre-treatment (baseline) samples for all relevant analyses.We are pleased to report that the final gene list obtained from this reanalysis remains consistent with our original findings derived from the full dataset. This outcome not only indicates the relative stability of the gene signatures we initially identified but also validates the effectiveness of the corrective measures taken during the reanalysis, thereby enhancing the reliability and accuracy of our results.Once again, we deeply appreciate your attentive review and guidance. Your expert comments have been instrumental in further refining our study and improving the overall quality of our work.

3.We sincerely thank you for reviewing our manuscript and providing this valuable feedback. You rightly pointed out the lack of clarity in the description of our data analysis workflow and the confusion regarding the methods used—a critical issue with which we fully concur. We acknowledge that this was a significant oversight in our manuscript, which compromised the clarity and reproducibility of our methods, and we sincerely apologize for this error.Upon carefully re-examining our analysis code and records, we confirm that the limma method was in fact used for differential gene expression analysis of the microarray data in this study. The limma package is suitable for analyzing gene expression data generated by either microarray or RNA-seq technologies. The mention of "DESeq2" in the Methods section was an unfortunate typographical error; while DESeq2 is typically applied to RNA-seq data, it was not used in our study. We take full responsibility for this confusion between the two methodological names.We have now completely removed all incorrect references to "DESeq2" and have thoroughly revised the Methods section to accurately and clearly describe the analytical workflow.

4.We sincerely thank you for this important and insightful comment. You have rightly pointed out the lack of clarity in our methodological description, particularly regarding the need for tailored analytical workflows for different data types. We fully agree that precise description of analytical methods is crucial for research reproducibility.Upon careful re-examination, we confirm that the limma package was consistently used for all differential expression analyses during the validation phase, including the processing of both microarray and RNA-seq datasets. The limma package is well-suited for analyzing gene expression data generated by either microarray or RNA-seq technologies, which ensured methodological consistency throughout our study.

5.We sincerely thank you for this important and expert comment. You correctly highlighted the risk that batch effect correction may potentially remove genuine biological signals, as well as the confounding influence of cellular composition on result interpretation. We fully agree that these methodological considerations are crucial for ensuring the reliability of our findings.Following your suggestion, we performed comprehensive PCA visualization analysis. We have now included PCA plots grouped by "Case/Control" status alongside those grouped by "Experimental Batch". These results clearly demonstrate that ComBat correction effectively removed technical variations across batches while successfully preserving and even highlighting the biological differences between the case group (triangles) and the control group (circles).Furthermore, we employed the CIBERSORT algorithm with its LM22 signature matrix to accurately estimate the proportions of 22 immune cell subtypes within each sample.

6.Thank you for pointing out this key terminology error. You are absolutely correct. In our study, we employed the Maximal Clique Centrality (MCC) method from the CytoHubba plugin to identify hub nodes within the network. We sincerely apologize for the typographical error in our original description, where it was mistakenly referred to as "modularity class," which is indeed inaccurate. This error has been comprehensively corrected throughout the revised manuscript.

7.Thank you for your attention to our analytical methodology. We selected a correlation threshold of 0.4 for network construction, which represents a widely accepted and moderately stringent cutoff in the field. This choice was based on the need to balance network scale with connection specificity: an excessively low threshold (e.g., 0.2) would introduce numerous weak or false-positive connections, resulting in an overly dense network, while an overly high threshold (e.g., 0.6) could exclude biologically meaningful signals, leading to an excessively sparse network. The threshold of 0.4 was chosen to retain meaningful biological associations while minimizing biases associated with overly lenient or stringent cutoffs. This threshold is further supported by its application in several published studies, such as:(Zhang R, Lan X, Zhu W, Wang L, Liu P, Li P. Regulation of autophagy by the PI3K-AKT pathway in Astragalus membranaceus–Cornus officinalis to ameliorate diabetic nephropathy. Front Pharmacol. 2025;16:1505637. Liu H, Wu M, Qi G, Ma F, Cao Y. Identification and Validation of Potential Diagnostic Biomarkers for Pulmonary Arterial Hypertension Based on Gene Expression Profiling. Pulm Circ. 2025;15(4):e70207.) To evaluate the robustness of the identified hub genes, we systematically varied the correlation threshold from 0.3 to 0.5 and re-identified hub genes under each condition. The results demonstrated that the top-ranked core hub genes remained largely stable across this range of thresholds. This confirms that these core hubs are not artifacts of a specific threshold value but represent intrinsic key nodes within the network structure.

Understanding the subcellular localization of hub genes is essential for interpreting their biological functions. As suggested, we performed a detailed compartment-based analysis using the Human Protein Atlas (HPA) database, which yielded important insights: the five hub genes exhibit a clear localization dichotomy, with 80% (SELL, CCR7, CD19, and CXCR4) being membrane-localized proteins primarily positioned on the plasma membrane, while the remaining 20% (CXCL13) is a secreted protein localized to the extracellular space. Based on these localization patterns, we propose a plausible functional divergence: the membrane proteins are likely involved in immune recognition and recruitment through direct cell-surface interactions, whereas the secreted protein CXCL13 may modulate the microenvironment via paracrine signaling. This spatial perspective provides a compelling explanation for their distinct immune-correlation patterns.

Tissue heterogeneity and the confounding effects of immune cell infiltration are critical issues that must be seriously considered. To directly quantify and address this problem, we systematically performed immune infiltration association analysis on the identified hub genes using the CIBERSORT algorithm. This analysis provided key evidence to distinguish whether changes in gene expression originate from differences in immune cell composition or from pathological programs of tissue-resident cells.

Our analysis yielded clear and discriminative conclusions: In UC, we observed significant heterogeneity in immune infiltration. Through correlation analysis, we successfully categorized the hub genes into two groups: one group (e.g., CD19, CXCL13) showed strong correlations with the abundance of adaptive immune cells (such as B cells and follicular helper T cells), indicating that their hub status is partially driven by immune cell infiltration; whereas the other group (e.g., CXCR4) remained relatively independent of immune cell variations, suggesting that it is more likely to reflect intrinsic pathological processes of intestinal resident cells. In RA, we discovered a distinctly different pattern. Immune infiltration in synovial tissue showed high consistency across samples, and more importantly, none of the hub genes (SELL, CCR7, CD19, CXCL13, CXCR4) showed significant correlations with the proportions of any immune cell type. This key evidence strongly indicates that in RA, the expression changes of our identified hub genes are not confounded by variations in immune cell composition among samples, but rather represent core pathogenic molecular programs that are stably activated within resident synovial cells during disease progression.We have systematically integrated these key analyses and conclusions into the manuscript and included the immune cell correlation results in the supplementary files. We once again thank you for your insightful comments, which have helped us better highlight the novelty and rigor of our study. We hope the current revisions meet your requirements.

8.We sincerely thank you for these rigorous and insightful comments. Your request for detailed statistical reporting and robustness validation is crucial for ensuring the reliability of our findings. In accordance with your suggestions, we have conducted comprehensive supplementary analyses, as detailed below: We have calculated and plotted ROC curves for each validated gene (Figure 10), and reported the log2FC (with 95% CI) and Benjamini-Hochberg (BH) adjusted p-values. These results have been compiled in a supplementary table included in the supplementary files. Furthermore, we rigorously performed a "leave-one-dataset-out" study-level validation. The results demonstrate that our identified hub genes remained statistically significant across all validation rounds, with stable AUC values, confirming the exceptional robustness of our findings.

9.Sincerely thank you for this rigorous and highly constructive feedback. You correctly pointed out that, in the absence of cross-cohort co-expression consistency validation, ceRNA analysis based on bioinformatic predictions is more appropriately framed as hypothesis-generating rather than as definitive evidence. We fully agree with this perspective. To ensure the robustness of our paper’s conclusions and adhere to the highest standards of scientific rigor, we have moved the ceRNA network figure from the main text to the supplementary materials. In both the Results and Methods sections of the manuscript, we have revised all relevant descriptions to clearly emphasize the exploratory and hypothesis-generating nature of this analysis, and have removed any language that could be interpreted as implying causal relationships or overstating the findings.

10.We sincerely thank you for this important comment. We fully agree that directly inferring "potential therapeutic targets" based on cross-sectional gene expression data and network centrality analysis constitutes an overinterpretation of the clinical implications and incorrectly implies causality. We have thoroughly revised the Discussion section to address this issue.

11.We thank the reviewer for this suggestion, which helps enhance the reproducibility of our work. In accordance with this request, we have included all requested materials in the supplementary files.

We again thank the reviewer for their valuable comments and look forward to your further feedback on our revised manuscript.

Best regards,

Wei-rong Cui

---

## [Editor Report · Decision Letter 2]

7 Dec 2025

Analysis of Common Differential Gene Expression between Rheumatoid Arthritis and Ulcerative Colitis

PONE-D-24-53954R2

Dear Dr. %Han%,

We’re pleased to inform you that your manuscript has been judged scientifically suitable for publication and will be formally accepted for publication once it meets all outstanding technical requirements.

Kind regards,

Srinivas Mummidi, D.V.M., Ph.D.

Academic Editor

PLOS One
---

## [Editor Report · Acceptance letter]

PONE-D-24-53954R2

PLOS One

Dear Dr. han,

I'm pleased to inform you that your manuscript has been deemed suitable for publication in PLOS One. Congratulations! Your manuscript is now being handed over to our production team.

Kind regards,

on behalf of

Prof Srinivas Mummidi

Academic Editor

PLOS One